# A CONSTRAINED BI-LEVEL OPTIMIZATION FRAME-WORK FOR CONSTRAINED REINFORCEMENT LEARNING FROM HUMAN FEEDBACK

## ABSTRACT

This paper studies the problem of jointly learning a reward function, a cost function, and a policy from human feedback. We formulate the problem as a constrained bi-level optimization problem, where the upper level infers the reward and cost functions from feedback, while the lower level optimizes a policy to best align with that feedback. To solve this problem, we propose a double-loop algorithm, Constrained Bi-level Optimization for Reinforcement Learning from Human Feedback (CB-RLHF), which solves the lower-level optimization problem in the inner loop and the upper-level optimization problem in the outer loop. We establish a theoretical guarantee that CB-RLHF converges at a rate of $\mathcal{O}(1/\sqrt{K})$, and we demonstrate its effectiveness across multiple simulation environments.

## 1 INTRODUCTION

In reinforcement learning (RL), an agent aims to learn an optimal policy that maximizes a cumulative reward in a dynamic environment. Designing an appropriate reward function is crucial to ensure that the learned policy aligns with the desired objective. However, reward engineering is challenging due to the need to capture numerous attributes (Knox et al., 2023) and the risk of reward hacking (Skalse et al., 2022). Reinforcement learning from human feedback (RLHF) has emerged as a promising approach to address these challenges. Existing RLHF algorithms (Christiano et al., 2017; Ouyang et al., 2022; Novoseller et al., 2020; Chen et al., 2022; Xu et al., 2020; Wu & Sun, 2023; Zhu et al., 2023; Kong & Yang, 2022; Zhan et al., 2023) typically follow a three-step iterative process: data collection, reward learning, and policy learning. First, humans provide feedback on data generated by the current policy. Second, a reward function is inferred from this feedback, often using a likelihood-based method grounded in the Bradley–Terry model (Bradley & Terry, 1952). Third, the agent updates its policy to maximize the cumulative return under the learned reward. These steps are repeated as the updated policy generates new data. By learning a reward function from human feedback, RLHF removes the need for manual reward design. For example, in natural language generation, human preference comparisons over candidate responses replace the impractical task of manually defining rewards for all possible outputs. RLHF has recently found success across diverse domains, including robotics (Hejna III & Sadigh, 2023), natural language processing (Ouyang et al., 2022), and computer games (Ibarz et al., 2018).

Despite recent progress, existing RLHF algorithms face two key limitations. The first is constraint inference. In many real-world scenarios, constraints are intrinsic, yet one-dimensional human feedback is insufficient to distinguish between objectives and constraints. Consider the case where a user poses a question that violates ethical guidelines. An ideal large language model should refuse to answer such a question to satisfy safety constraints. However, from the standpoint of helpfulness, the question remains unaddressed, and the response becomes unhelpful. As discussed in (Dai et al., 2023), the inherently conflicting factors (safety and helpfulness) arise when humans provide preference feedback. In these situations, it would be preferable for humans to indicate preferences across multiple dimensions, such as safety and helpfulness, rather than being restricted to a single axis of comparison. Otherwise, ambiguity arises, since one-dimensional feedback cannot fully express intentions over competing objectives. Meanwhile, the single reward function learned from one-dimensional preference feedback may be insufficient to capture the full spectrum of these conflict-

ing factors. As a result, existing RLHF methods can produce policies that violate these constraints, leading to undesired or unsafe behaviors.

The second limitation is misalignment (Chakraborty et al., 2023), which occurs when the learned reward function fails to fully capture the human's intended objective, leading to suboptimal policies. In most RLHF methods, feedback collection, reward learning, and policy optimization are performed iteratively. As a result, the dependence of the learned reward function on feedback generated by the evolving policy is overlooked. Some RLHF algorithms (Lee et al., 2021; Park et al., 2022) typically update the reward function using an expanding dataset. In each iteration, new trajectories from the current policy are added to the dataset. However, the reward function is updated using batches of trajectories collected in previous iterations, without incorporating feedback on trajectories generated by the current optimal policy. This disconnect allows agents to exploit inaccuracies in the reward function and gradually drift away from the intended behavior, thereby exacerbating misalignment.

**Contribution.** The contributions of this paper are fourfold. First, we study the constrained RLHF problem where human feedback reflects both optimality and constraint satisfaction. We formulate this problem as a constrained bi-level optimization problem. In particular, the upper level aims to learn both a reward function and a cost function from human feedback. The lower level aims to learn a policy that optimizes the learned reward while satisfying the learned cost. The general approach is to treat the lower-level optimization problem as a constrained RL task. However, the constrained RL problem is inherently non-convex, making hypergradient (i.e., the gradient of the upper-level optimization problem) computation possibly intractable since it requires the inverse Hessian of the lower-level objective function. To overcome this challenge, we employ its dual problem to reformulate the lower-level optimization problem. This reformulation guarantees the existence of the required inverse Hessian while preserving the result policy equivalence to the original constrained RL formulation. In this framework, the upper- and lower-level decision variables are coupled: human feedback used to optimize the reward and cost functions is generated from the data induced by the optimal policy according to the current reward and cost functions. This dependency resolves the misalignment limitation. Second, we propose Constrained Bi-level optimization for Reinforcement Learning from Human Feedback (CB-RLHF), an algorithm for solving this constrained bi-level optimization problem. A particular challenge is to compute the hypergradient because it includes the derivative of the lower-level optimal solution, which is possibly non-differentiable. We overcome this by applying the Clarke subdifferential approximation with two gradient evaluations. Third, we provide theoretical guarantees showing that the algorithm converges at a rate of $\mathcal{O}(1/\sqrt{K})$, and we validate the performance gap between the optimal policy and the learned policy. Fourth, we demonstrate the effectiveness of our approach through four experiments, showing that CB-RLHF simultaneously resolves the misalignment and constraint inference limitations at once.

## 2 RELATED WORK

**RLHF.** Human feedback in RLHF generally falls into two major classes: ranking and pairwise comparison. Pairwise comparisons can be further divided into two types. The first is action-level preference (Zhu et al., 2023; Kong & Yang, 2022; Zhan et al., 2023; Li et al., 2023), where humans indicate which action is preferable given a particular state. The second is trajectory-level preference (Novoseller et al., 2020; Chen et al., 2022; Xu et al., 2020; Wu & Sun, 2023), where humans specify which of two trajectories is preferred. In this paper, we adopt trajectory-level preference. Existing RLHF methods can be broadly categorized into online and offline variants. Online approaches (Novoseller et al., 2020; Saha et al., 2023; Chen et al., 2022; Xu et al., 2020; Wu & Sun, 2023; Dai et al., 2023) continuously update the reward model and policy through interactive agent-environment interactions, where human feedback is collected and incorporated during training. This enables iterative refinement of both the reward function and the agent's policy, improving alignment and mitigating overoptimization. Recent works (Chakraborty et al., 2023; Shen et al., 2024) formulate RLHF as a bi-level optimization problem. In (Chakraborty et al., 2023), the upper-level problem learns a reward function from human feedback, while the lower-level problem computes the optimal policy with respect to the learned reward. Meanwhile, (Shen et al., 2024) replaces the lower-level optimization problem with a penalty-based formulation, integrating it directly into the upper-level objective function. Safe RLHF Dai et al. (2023) decouples human feedback into two components: helpfulness and harmlessness. This formulation yields unbiased preference signals while balancing two potentially conflicting objectives. In particular, the harmlessness objective can be naturally in-

terpreted as a constraint in the optimization problem. Offline approaches (Zhu et al., 2023; Zhan et al., 2023; Li et al., 2023) train the reward model and policy on a pre-collected dataset and corresponding human feedback. This approach is well-suited for domains where online interaction is costly, unsafe, or impractical.

**Constrained bi-level optimization.** There has been growing interest in constrained bi-level optimization problems, where the lower-level decision variable is subject to constraints. Such problems arise across diverse domains, including machine learning (Xu & Zhu, 2023b), robotics (Ming et al., 2023), and economics (Mehdipourpicha & Bo, 2020). Two primary approaches have been proposed to address constrained bi-level optimization problems: value function-based methods (Ye & Zhu, 2010; Liu et al., 2023; Yao et al., 2024) and implicit gradient-based methods (Gould et al., 2016; Tsaknakis et al., 2022; Khanduri et al., 2023; Xiao et al., 2023; Xu & Zhu, 2023a). The value function-based approach reformulates the original bi-level problem into a single-level problem using the value function. The implicit gradient-based approach solves the constrained lower-level problem using the Lagrangian framework and derives gradients via implicit differentiation. However, a key limitation of both approaches is the assumption that the lower-level constraints are convex. In this paper, we relax this convexity assumption.

## 3 MODEL AND PROBLEM STATEMENT

In this section, we introduce our formulation of the constrained RLHF problem.

**Markov Decision Process (MDP).** Consider a MDP without a reward function, denoted as $MDP\backslash R$ $(S, A, \gamma, H, P_0, P)$, which includes the state space $S$, the action space $A$, the discount factor $\gamma \in (0, 1]$, the finite horizon $H \in \mathbb{Z}^+$ and the initial state distribution $P_0$. The transition function $P(s'|s, a)$ denotes the probability density of transitioning from state $s \in S$ to state $s' \in S$ upon taking action $a \in A$. A policy $\pi(a|s)$ specifies the probability density of selecting action $a$ given state $s$. A trajectory $\tau$ is defined as a sequence of state-action pairs over the horizon, i.e., $\tau = s_0, a_0, s_1, a_1, \ldots, s_{H-1}, a_{H-1}$.

**Human Feedback.** Given any two trajectories, a human could provide three types of feedback (Dai et al., 2023): (i) preference label $y_r \in \{0, 1\}$ comparing optimality, (ii) preference label $y_c \in \{0, 1\}$ comparing constraint satisfaction, and (iii) classification label $z \in \{1, -1\}$ indicating whether a trajectory violates the constraints. For any trajectory pair $(\tau_0, \tau_1)$, the label $y_r = 0$ if the human prefers $\tau_0$ over $\tau_1$ in terms of optimality; otherwise, $y_r = 1$. Similarly, the label $y_c = 1$ if the human prefers $\tau_0$ over $\tau_1$ in terms of constraint satisfaction; otherwise, $y_c = 0$. For any trajectory $\tau$, the label $z = 1$ if $\tau$ violates the constraints and $z = -1$ otherwise. We define the unknown human feedback distribution as $h_l(y_r, y_c, z_0, z_1|\tau_0, \tau_1)$ where $z_0$ and $z_1$ are the classification labels for trajectories $\tau_0$ and $\tau_1$, respectively.

**Problem statement.** An agent aims to learn a policy $\pi^*$ that best aligns with human feedback. Consider a trajectory $\tau_0$ generated by the policy $\pi^*$ and a trajectory $\tau_1$ generated by any other policy $\pi'$. By definition, a human is expected to prefer $\tau_0$ over $\tau_1$ both in terms of optimality and constraint satisfaction, and to regard $\tau_0$ as not violating any constraints ($y_r = 0, y_c = 1$ and $z = -1$ for $\tau_0$).

## 4 PROBLEM FORMULATION AND CONSTRAINED BI-LEVEL SETUP

To learn the policy $\pi^*$, the agent employs two parametric functions: a reward function $r_\phi : S \times A \to \mathbb{R}$ and a cost function $c_\psi : S \times A \to \mathbb{R}$, parameterized by decision variables $\phi \in \mathbb{R}^{d_\phi}$ and $\psi \in \mathbb{R}^{d_\psi}$, respectively. To mitigate the misalignment issue, the agent requires human feedback on trajectories generated by the optimal policy corresponding to the current reward and cost functions. The agent thus simultaneously learns the reward function parameter $\phi$, the cost function parameter $\psi$, and the associated policy. We formulate this problem as a constrained bi-level optimization problem where the lower-level optimization problem learns a policy via constrained RL under the current $r_\phi$ and $c_\psi$ and the upper-level optimization problem updates $r_\phi$ and $c_\psi$ by minimizing a cross-entropy loss with human feedback.

**The lower-level optimization problem.** Define the cumulative reward of a policy $\pi$ under reward function $r_\phi$ as $J_{r_\phi}(\pi) \triangleq E^\pi[\sum_{h=0}^{H-1} \gamma^h r_\phi(s_h, a_h)]$, where the initial state is sampled from the

distribution $P_0$. Analogously, define the cumulative cost as $J_{c_\psi}(\pi) \triangleq E^\pi[\sum_{h=0}^{H-1} \gamma^h c_\psi(s_h, a_h)]$. Given $r_\phi$ and $c_\psi$, the agent aims to solve the following constrained RL problem:

$$\max_\pi \quad J_{r_\phi}(\pi) + H(\pi), \quad \text{s.t.} \quad J_{c_\psi}(\pi) \leq 0, \tag{1}$$

where $H(\pi) \triangleq E^\pi[\sum_{h=0}^{H-1} -\gamma^h \ln(\pi(a_h|s_h))]$ is the discounted entropy. The problem (1) is non-convex because both the objective function and the constraint are non-convex. Since the primal problem (1) is difficult to solve due to its non-convexity, we instead derive its dual formulation:

$$\min_{\lambda \geq 0} G(\lambda; \phi, \psi), \tag{2}$$

where the dual function is defined as $G(\lambda; \phi, \psi) \triangleq \max_\pi J_{r_\phi}(\pi) + H(\pi) - \lambda J_{c_\psi}(\pi)$ and $\lambda$ is the Lagrange multiplier. Notably, the dual problem (2) is a convex optimization problem. Let $\lambda^*(\phi, \psi)$ denote the optimal solution to the dual problem (2). With $\lambda^*(\phi, \psi)$ we can get the policy $\pi_{\phi,\psi,\lambda^*(\phi,\psi)} = \arg\max_\pi J_{r_\phi}(\pi) + H(\pi) - \lambda^*(\phi, \psi) J_{c_\psi}(\pi)$. As shown in the following lemma, this policy solves the primal constrained RL problem (1).

**Lemma 1.** *The strong duality holds for the primal problem (1) and its dual problem (2). The policy $\pi_{\phi,\psi,\lambda^*(\phi,\psi)}$ is identical to the optimal policy of the primal problem (1).*

The proof of Lemma 1 is provided in Appendix A.3. Since the policy $\pi_{\phi,\psi,\lambda^*(\phi,\psi)}$ coincides with the optimal solution to the primal problem (1), we adopt the dual formulation (2) as the lower-level optimization problem. This allows us to compute both the dual optimal solution $\lambda^*(\phi, \psi)$ and the corresponding policy $\pi_{\phi,\psi,\lambda^*(\phi,\psi)}$.

**The upper-level optimization problem.** With the policy $\pi_{\phi,\psi,\lambda^*(\phi,\psi)}$ obtained from the lower-level optimization problem, the agent samples trajectories to elicit human feedback. To model preferences, we adopt the Bradley–Terry model, which predicts human choices based on the current reward and cost functions. For any trajectory $\tau$, define the cumulative reward of this trajectory as $J_{r_\phi}(\tau) \triangleq \sum_{h=0}^{H-1} \gamma^h r_\phi(s_h, a_h), s_h, a_h \in \tau$ and analogously, the cumulative cost of this trajectory is $J_{c_\psi}(\tau) \triangleq \sum_{h=0}^{H-1} \gamma^h c_\psi(s_h, a_h), s_h, a_h \in \tau$. Given a trajectory pair $\tau_0, \tau_1$, the probabilities that a human prefers $\tau_0$ over $\tau_1$ in terms of optimality and constraint satisfaction are estimated by the Bradley–Terry model as follows:

$$P_{r_\phi}(\tau_0 \succ \tau_1) = \frac{\exp(J_{r_\phi}(\tau_0))}{\exp(J_{r_\phi}(\tau_0)) + \exp(J_{r_\phi}(\tau_1))} = \sigma(J_{r_\phi}(\tau_0) - J_{r_\phi}(\tau_1)) \text{ for optimality,}$$

$$P_{c_\psi}(\tau_0 \succ \tau_1) = \frac{\exp(J_{c_\psi}(\tau_0))}{\exp(J_{c_\psi}(\tau_0)) + \exp(J_{c_\psi}(\tau_1))} = \sigma(J_{c_\psi}(\tau_0) - J_{c_\psi}(\tau_1)) \text{ for constraint satisfaction,}$$

where $\sigma(x) \triangleq \frac{1}{1+\exp(-x)}$.

The agent employs the reward function $r_\phi$ to model the human preferences over optimality, and the cost function $c_\psi$ to model both the constraint satisfaction and constraint violation classification. Using human feedback together with the Bradley–Terry model, the parameters $\phi$ and $\psi$ are learned by maximizing the log-likelihood of the observed labels.

The pairwise comparison loss for the reward parameter $\phi$, based on the optimality preference label $y_r$, is defined as:

$$L_r(\phi, \lambda^*(\phi, \psi)) \triangleq - E_{(\tau_0,\tau_1,y_r) \sim D(y_r,y_c,z_0,z_1,\tau_0,\tau_1;\pi_{\phi,\psi,\lambda^*(\phi,\psi)})}[(1 - y_r) \log \sigma(J_{r_\phi}(\tau_0) - J_{r_\phi}(\tau_1))$$
$$+ y_r \log \sigma(J_{r_\phi}(\tau_1) - J_{r_\phi}(\tau_0))]. \tag{3}$$

The sample distribution of trajectories and the collected human feedback is $D(y_r, y_c, z_0, z_1, \tau_0, \tau_1; \pi_{\phi,\psi,\lambda^*(\phi,\psi)}) \triangleq h_l(y_r, y_c, s_0, s_1|\tau_0, \tau_1)\rho(\tau_0; \pi_{\phi,\psi,\lambda^*(\phi,\psi)})\rho(\tau_1; \pi_{\phi,\psi,\lambda^*(\phi,\psi)})$, where $\rho(\tau; \pi_{\phi,\psi,\lambda^*(\phi,\psi)}) \triangleq P_0(s_0)\Pi_{h=0}^{H-1} P(s_{h+1}|s_h, a_h)\pi_{\phi,\psi,\lambda^*(\phi,\psi)}(a_h|s_h), s_h, a_h \in \tau$ is the trajectory distribution induced by the policy $\pi_{\phi,\psi,\lambda^*(\phi,\psi)}$. Notice that $h_l(y_r, y_c, z_0, z_1|\tau_0, \tau_1)$ reflects the human feedback distribution, which is realized through sampling. The loss function (3) represents the cross-entropy loss between the predictions based on $r_\phi$ and the actual human

preference label $y_r$. Specifically, when $y_r = 0$, the loss encourages the reward function $r_\phi$ to assign a higher cumulative reward to the trajectory $\tau_0$; conversely, when $y_r = 1$, it encourages a higher reward for $\tau_1$.

The total loss for the cost parameter $\psi$, incorporating both the constraint satisfaction preference label $y_c$ and the classification label $z$, is defined as:

$$
\begin{aligned}
&L_c(\psi, \lambda^*(\phi, \psi)) \\
&= -E_{(\tau_0, \tau_1, y_c, z_0, z_1) \sim D(y_r, y_c, z_0, z_1, \tau_0, \tau_1; \pi_{\phi, \psi, \lambda^*(\phi, \psi)})} [\underbrace{(1 - y_c) \log \sigma(J_{c_\psi}(\tau_0) - J_{c_\psi}(\tau_1))}_{\text{part 1}} \\
&\underbrace{+ y_c \log \sigma(J_{c_\psi}(\tau_1) - J_{c_\psi}(\tau_0))}_{\text{part 1}} + \underbrace{\log \sigma(z_0 J_{c_\psi}(\tau_0)) + \log \sigma(z_1 J_{c_\psi}(\tau_1))}_{\text{part 2}}].
\end{aligned}
\tag{4}
$$

Part 1 corresponds to the pairwise comparison loss for the cost parameter $\psi$, which is similar to equation (3). When $y_c = 0$, the comparison loss encourages the cost function $c_\psi$ to assign a higher cumulative cost to the trajectory $\tau_0$, and when $y_r = 1$, it encourages a higher cost for $\tau_1$. In this way, the loss captures human preferences for trajectories with better constraint satisfaction. Part 2 leverages the additional classification labels $z_0, z_1$ to further refine the learning of the cost function. Assume the existence of a baseline trajectory $\tau_b$ such that $J_{c_\psi}(\tau_b) = 0$. If a trajectory $\tau$ violates the constraint (i.e., $z = 1$), the cost function $c_\psi$ should assign a higher cost to $\tau$ and $P_{c_\psi}(\tau \succ \tau_b) = \sigma(J_{c_\psi}(\tau) - J_{c_\psi}(\tau_b)) = \sigma(z J_{c_\psi}(\tau))$. Conversely, if a trajectory $\tau$ does not violate the constraint (i.e., $z = -1$), the baseline trajectory $\tau_b$ should be assigned a higher cost and $P_{c_\psi}(\tau_b \succ \tau) = \sigma(J_{c_\psi}(\tau_b) - J_{c_\psi}(\tau)) = \sigma(z J_{c_\psi}(\tau))$ (Dai et al., 2023).

The agent aims to find the parameters $\phi$ and $\psi$ that minimize the loss functions defined in Equations (3) and (4). This leads to the following upper-level optimization problem:

$$
\arg\min_{\phi, \psi} F(\phi, \psi, \lambda^*(\phi, \psi)) \triangleq L_r(\phi, \lambda^*(\phi, \psi)) + L_c(\psi, \lambda^*(\phi, \psi)).
\tag{5}
$$

**Constrained bi-level optimization formulation.** By coupling the upper-level optimization problem (5) with the lower-level optimization problem (2), we obtain the following constrained bi-level optimization problem:

$$
\arg\min_{\phi, \psi} F(\phi, \psi, \lambda^*(\phi, \psi)), \text{ s.t. } \lambda^*(\phi, \psi) = \arg\min_{\lambda \geq 0} \quad G(\lambda; \phi, \psi).
\tag{6}
$$

The lower-level problem computes the optimal Lagrange multiplier $\lambda^*(\phi, \psi)$ and corresponding optimal policy $\pi_{\phi, \psi, \lambda^*(\phi, \psi)}$ based on the given reward function $r_\phi$ and cost function $c_\psi$. The upper-level problem then learns the reward function $r_{\phi^*}$ and the cost function $c_{\psi^*}$ by minimizing the losses derived from human feedback, which incorporate optimality preferences, constraint satisfaction preferences, and classification labels.

## 5 ALGORITHM

In this section, we develop the algorithm CB-RLHF to solve the constrained bi-level optimization problem (6). A key challenge lies in computing the hypergradients $H_\phi(\phi, \psi, \lambda^*(\phi, \psi))$ and $H_\psi(\phi, \psi, \lambda^*(\phi, \psi))$, both of which depend on the lower-level optimal solution $\lambda^*(\phi, \psi)$. The expressions for these two hypergradients are

$$
H_\phi(\phi, \psi, \lambda^*(\phi, \psi)) = \nabla_\phi F(\phi, \psi, \lambda^*(\phi, \psi)) + \nabla_\phi \lambda^*(\phi, \psi)^T \nabla_\lambda F(\phi, \psi, \lambda^*(\phi, \psi)),
\tag{7}
$$

and

$$
H_\psi(\phi, \psi, \lambda^*(\phi, \psi)) = \nabla_\psi F(\phi, \psi, \lambda^*(\phi, \psi)) + \nabla_\psi \lambda^*(\phi, \psi)^T \nabla_\lambda F(\phi, \psi, \lambda^*(\phi, \psi)).
\tag{8}
$$

To compute these hypergradients, the upper-level objective function $F(\phi, \psi, \lambda^*(\phi, \psi))$ and the mapping $\lambda^*(\phi, \psi)$ need to be continuously differentiable with respect to $\phi$ and $\psi$. However, this assumption is restrictive as the lower-level optimization problem in (6) is constrained and $\lambda^*(\phi, \psi)$ may fail to be differentiable (Xu & Zhu, 2023a). In particular, given any $\phi'$, if there is a non-differentiable point in its neighborhood $\mathcal{B}(\phi', \epsilon)$, a ball centered at $\phi'$ with radius $\epsilon$, the gradient descent may break

down. In such cases, the updates of $\phi'$ become non-smooth around the non-differentiable point, as the hypergradient is not well-defined there. This non-differentiability presents a fundamental obstacle for both hypergradient computation and algorithm design.

We adopt the Clarke subdifferential framework (Clarke, 1975) to overcome this challenge.

**Definition 1.** *The Clarke subdifferential (Clarke, 1975) of a locally Lipschitz function $f(x) : \mathbb{R}^n \to \mathbb{R}$ at a point $x$ is defined as $\bar{\partial} f(x) \triangleq conv\{s \in \mathbb{R}^n : \exists \{x_i\}_{i=1}^{\infty} s.t. x_i \to x, \nabla f(x_i) \to s\}$.*

Following Definition 1, the subdifferential with respect to $\phi$ can be approximated by extending the Clarke subdifferential as $conv\{H_\phi(\phi', \psi, \lambda^*(\phi', \psi)) | \phi' \in \mathcal{B}(\phi, \epsilon)\}$. For any given $\phi$, we can sample all feasible points in the neighborhood $\mathcal{B}(\phi, \epsilon)$, compute the corresponding gradients, and take their convex hull to approximate the subdifferential. If all points in $\mathcal{B}(\phi, \epsilon)$ are differentiable, then the convex hull coincides with the exact hypergradient. Otherwise, the approximation accounts for the influence of non-differentiable points. Analogously, the approximated subdifferential with respect to $\psi$ is $conv\{H_\psi(\phi, \psi', \lambda^*(\phi, \psi')) | \psi' \in \mathcal{B}(\psi, \epsilon)\}$. These approximated subdifferentials can then be used to update the decision variables, serving as a substitute for gradient descent with exact hypergradients.

However, computing a hypergradient is computationally expensive. This is primarily because it typically involves calculating the second-order Hessian of the lower-level objective function, which is prohibitively costly to compute. Moreover, a large number of sampled points further amplifies the computational burden, as a hypergradient needs to be computed at each sampled neighbor. As a result, approximating the hypergradient for all sampled neighbors significantly increases the overall computational cost.

We address this computational challenge in two ways. First, for each $\phi$ or $\psi$, we approximate the hypergradient when a non-differentiable point is in the neighborhood $\mathcal{B}(\phi, \epsilon)$ or $\mathcal{B}(\psi, \epsilon)$; otherwise, we compute the exact hypergradient. This selective approach reduces computational cost by avoiding unnecessary approximations. Second, inspired by Equation (5) in (Xu & Zhu, 2023a), we further approximate the hypergradient using only two gradient evaluations, thereby achieving a tractable and efficient computation.

## 5.1 HYPERGRADIENT COMPUTATION

When $\lambda^*(\phi, \psi)$ is continuously differentiable at $\phi$ or $\psi$, we compute $H_\phi(\phi, \psi, \lambda^*(\phi, \psi))$ or $H_\psi(\phi, \psi, \lambda^*(\phi, \psi))$. Otherwise, we approximate the subdifferential at $\phi$ or $\psi$.

**Check non-differentiability.** Since the upper-level objective function $F(\phi, \psi, \lambda^*(\phi, \psi))$ in the problem (6) is continuously differentiable (as shown in Appendix A.7), whereas the lower-level optimal solution $\lambda^*(\phi, \psi)$ may not be. Therefore, it is necessary to verify the differentiability of $\lambda^*(\phi, \psi)$. To this end, we consider the Lagrangian formulation of the convex optimization problem (2) as $L(\phi, \psi, \lambda, \nu) \triangleq G(\lambda; \phi, \psi) - \nu(\phi, \psi) \lambda(\phi, \psi)$, where $\nu(\phi, \psi)$ denotes the Lagrangian multiplier. The procedure for checking differentiability with respect to each $\phi$ on $\mathcal{B}(\phi', \epsilon)$ is summarized in Proposition 1, and the process for each $\psi$ is analogous.

**Proposition 1.** *For any $\phi$ and $\epsilon > 0$, the lower-level optimal solution $\lambda^*(\phi, \psi)$ is continuously differentiable on $\mathcal{B}(\phi, \epsilon)$ if there exists $\delta > 0$ that $\nu(\phi, \psi) > \|\nabla_\phi \nu(\phi, \psi)\| \epsilon$ or $\lambda^*(\phi, \psi) > \|\nabla_\phi \lambda^*(\phi, \psi)\| \epsilon$.*

The explicit expressions of $\nu(\phi, \psi)$ and $\lambda^*(\phi, \psi)$ are shown in Appendix A.4. When either inequality is satisfied, no point in the neighborhood $\mathcal{B}(\phi, \epsilon)$ results in $\nu(\phi, \psi) = 0$ and $\lambda^*(\phi, \psi) = 0$. When $\nu(\phi, \psi) = 0$ and $\lambda^*(\phi, \psi) = 0$, the lower-level optimal solution $\lambda^*(\phi, \psi)$ lies exactly on the boundary of the constraint, while the constraint itself does not affect the optimality condition. This creates a fragile balance where even small perturbations in $\phi$ can abruptly alter whether the constraint is active in shaping $\lambda^*(\phi, \psi)$. Due to the abrupt nature of this transition, the mapping from $\phi$ to $\lambda^*(\phi, \psi)$ becomes non-smooth and thus non-differentiable.

**Computing exact hypergradients.** When the conditions in Proposition 1 are satisfied, the exact hypergradients can be directly computed. In this case, we obtain the hypergradients $H_\phi(\phi, \psi, \lambda^*(\phi, \psi))$ and $H_\psi(\phi, \psi, \lambda^*(\phi, \psi))$ from Equation (7) and (8), respectively. These hypergradients are computed by applying the chain rule to differentiate the upper-level objective function $F(\phi, \psi, \lambda^*(\phi, \psi))$ with respect to $\phi$ and $\psi$.

**Approximating subdifferentials.** When the conditions in Proposition 1 are violated, the corresponding subdifferentials are approximated as follows:

$$\mathcal{S}_\phi = \{\kappa H_\phi(\phi, \psi, \lambda^*(\phi, \psi)) + (1 - \kappa)\nabla_\phi F(\phi, \psi, \lambda^*(\phi, \psi)) | \kappa \in [0, 1]\} \quad (9)$$

and

$$\mathcal{S}_\psi = \{\kappa H_\psi(\phi, \psi, \lambda^*(\phi, \psi)) + (1 - \kappa)\nabla_\psi F(\phi, \psi, \lambda^*(\phi, \psi)) | \kappa \in [0, 1]\}. \quad (10)$$

One gradient evaluation corresponds to the exact hypergradient $H_\phi(\phi, \psi, \lambda^*(\phi, \psi))$ or $H_\psi(\phi, \psi, \lambda^*(\phi, \psi))$, while the other is obtained by considering the lower-level optimal solution as zero which reduces $H_\phi(\phi, \psi, \lambda^*(\phi, \psi))$ to $\nabla_\phi F(\phi, \psi, \lambda^*(\phi, \psi))$ or $H_\psi(\phi, \psi, \lambda^*(\phi, \psi))$ to $\nabla_\psi F(\phi, \psi, \lambda^*(\phi, \psi))$. Since each subdifferential is approximated using two gradient evaluations, the convex hull simplifies to a linear combination. With this approximation strategy, we are now ready to present the complete algorithm for solving the constrained bi-level optimization problem.

---

**Algorithm 1** Constrained Bi-level optimization for RLHF (CB-RLHF)

---

1: Initialize parameters $\phi^0$ and $\psi^0$, step size sequences $\{\alpha_k\}, \{\beta_k\}, \{\omega_k\}$, and the initial radius $\epsilon_0$
2: **for** $k = 1, \cdots, K$ **do**
3:     $\epsilon_k \leftarrow \frac{1}{k^2}$
4:     **for** $t = 0, 1, \cdots, t_k - 1$ **do**
5:         Compute $\lambda^{t+1} = \max(0, \lambda^t - \omega_k \nabla_\lambda G(\lambda^t; \phi^k, \psi^k))$
6:     **end for**
7:     $\lambda^k(\phi^k, \psi^k) = \lambda^{t_k-1}(\phi^k, \psi^k)$
8:     **if** $\nu(\phi^k, \psi^k) > \|\nabla_\phi \nu(\phi^k, \psi^k)\|\epsilon^k$ or $\lambda^k(\phi^k, \psi^k) > \|\nabla_\phi \lambda^k(\phi^k, \psi^k)\|\epsilon^k$ **then**
9:         Compute $g_\phi^k = H_\phi(\phi^k, \psi^k, \lambda^k(\phi^k, \psi^k))$ through Equation (7)
10:     **else**
11:         Compute $g_\phi^k = \arg\min_{g \in S_\phi} \|g\|$
12:     **end if**
13:     **if** $\nu(\phi^k, \psi^k) > \|\nabla_\psi \nu(\phi^k, \psi^k)\|\epsilon^k$ or $\lambda^k(\phi^k, \psi^k) > \|\nabla_\psi \lambda^k(\phi^k, \psi^k)\|\epsilon^k$ **then**
14:         Compute $g_\psi^k = H_\psi(\phi^k, \psi^k, \lambda^k(\phi^k, \psi^k))$ through Equation (8)
15:     **else**
16:         Compute $g_\psi^k = \arg\min_{g \in S_\psi} \|g\|$
17:     **end if**
18:     $\phi^{k+1} \leftarrow \phi^k - \alpha_k g_\phi^k$
19:     $\psi^{k+1} \leftarrow \psi^k - \beta_k g_\psi^k$
20: **end for**

---

## 5.2 Algorithm statement

The CB-RLHF algorithm is summarized in Algorithm 1. At each iteration $k$, the agent first partially solves the lower-level optimization problem within an inner loop and then uses this intermediate solution to address the upper-level optimization problem in the outer loop. The next step is to determine whether to compute exact hypergradients or approximate subdifferentials by checking the conditions in Proposition 1. For each $\phi^k$, if $\lambda^k(\phi^k, \psi^k)$ is continuously differentiable over the neighborhood $\mathcal{B}(\phi^k, \epsilon)$, the hypergradient is computed directly by extending Equation (7). Otherwise, it is approximated using Equation (9). An analogous procedure is applied for $\psi^k$, using either Equation (8) or Equation (10). Finally, the parameters $\phi^k$ and $\psi^k$ are updated via gradient descent.

## 6 Analytical result

In this section, we present our analytical results on the convergence rate of the hypergradient and the performance gap between the learned policy and the human policy. To facilitate the analysis, we impose the following assumptions on the estimated reward function and the cost functions.

**Assumption 1.** *For any $s \in S, a \in A$, the reward function $r_\phi(s, a)$ satisfies $|r_\phi(s, a)| \leq R_{b1}, \|\nabla_\phi r_\phi(s, a)\| \leq R_{b2}$, and $\|\nabla_\phi^2 r_\phi(s, a)\| \leq R_{b3}$ for any $\phi$. Similarly, the cost function $c_\psi(s, a)$ satisfies $|c_\psi(s, a)| \leq C_{b1}, \|\nabla_\psi c_\psi(s, a)\| \leq C_{b2}$, and $\|\nabla_\psi^2 c_\psi(s, a)\| \leq C_{b3}$ for any $\psi$.*

**Assumption 2.** *We assume there exists a reward (cost) model class $\mathcal{F}$, the estimated reward function $r_\phi$, the estimated cost function $c_\psi$, the true reward function of the human $r_h$, the true cost function of the human $c_h$ belong to $\mathcal{F}$.*

Assumption 1 implies that the derivatives of the reward function $r_\phi$ and the cost function $c_\psi$ are bounded. Such boundedness of higher-order derivatives is a standard assumption and has been widely adopted in bi-level optimization (Liu & Zhu, 2022; Zeng et al., 2022), RL (Wang et al., 2019; Zhang et al., 2020), and RLHF (Zhu et al., 2023). Assumption 2 further states that all reward and cost functions belong to a common function class, such as neural networks. This assumption has also been employed in prior RLHF studies (Li et al., 2023).

Let $d(\mathcal{S}, \mathcal{S}') \triangleq \min\{\|x - y\| | x \in \mathcal{S}, y \in \mathcal{S}'\}$ denote the distance between two sets $\mathcal{S}$ and $\mathcal{S}'$. We now state Lemma 2, which establishes the feasibility of approximating the subdifferential using only two hypergradient evaluations.

**Lemma 2.** *Assume there exists a small $\epsilon > 0$ with a $\phi' \in B(\phi, \epsilon)$ leads to $\lambda^*(\phi', \psi)$ non-differentiable at $\phi'$. We have $d(S_\phi, conv\{H_\phi(\phi'', \psi, \lambda^*(\phi'', \psi)) | \phi'' \in \mathcal{B}(\phi, \epsilon), \lambda^*(\phi'', \psi) \text{ differentiable}\}) \leq \mathcal{O}(\epsilon)$.*

Lemma 2 shows that the approximation error between the estimated subdifferential $\mathcal{S}_\phi$ and $conv\{H_\phi(\phi'', \psi, \lambda^*(\phi'', \psi)) | \phi'' \in \mathcal{B}(\phi, \epsilon), \lambda^*(\phi'', \psi) \text{ differentiable}\}$ is bounded by $\mathcal{O}(\epsilon)$. An analogous result holds for $\psi$. Therefore, a convex combination of just two gradient evaluations provides a computationally efficient yet theoretically sound approximation of the subdifferential. This construction guarantees stable updates even in the presence of non-differentiable points. Accounting for this approximation error, we now establish the convergence rate of the proposed algorithm in the following theorem.

**Theorem 1.** *Suppose Assumption 1 holds, by choosing $\alpha_k, \beta_k \propto \frac{1}{\sqrt{K}}$ and $\epsilon = \frac{1}{k^2}$, the following convergence guarantee holds:*

$$\min\{\|g_\phi\|^2 | g_\phi \in S_\phi\} \leq \mathcal{O}(\frac{1}{\sqrt{K}}), \text{and } \min\{\|g_\psi\|^2 | g_\psi \in S_\psi\} \leq \mathcal{O}(\frac{1}{\sqrt{K}}).$$

Theorem 1 shows that Algorithm 1 converges at a rate of $\mathcal{O}(\frac{1}{\sqrt{K}})$. This result implies that the proposed subdifferential approximation method (Equation (9) and (10)) effectively approximates the true subdifferential.

Let $N(\mathcal{F}, \|\cdot\|_\infty, \eta)$ denote the $\eta-$covering number for the function class $\mathcal{F}$ under the infinity norm $\|\cdot\|_\infty$ and $n$ as the number of preference data collected at each iteration. We define the cumulative reward difference between $\pi_{\phi^*, \psi^*, \lambda^*(\phi^*, \psi^*)}$ and human policy $\pi_h$ according to the true reward function $r_h$ as $SubOptR(\pi_{\phi^*, \psi^*, \lambda^*(\phi^*, \psi^*)}) \triangleq J_{r_h}(\pi_h) - J_{r_h}(\pi_{\phi^*, \psi^*, \lambda^*(\phi^*, \psi^*)})$. Analogously, the cumulative cost difference between $\pi_{\phi^*, \psi^*, \lambda^*(\phi^*, \psi^*)}$ and $\pi_h$ according to the true cost function $c_h$ is defined as $SubOptC(\pi_{\phi^*, \psi^*, \lambda^*(\phi^*, \psi^*)}) \triangleq J_{c_h}(\pi_{}) - J_{c_h}(\pi_{\phi^*, \psi^*, \lambda^*(\phi^*, \psi^*)})$.

**Theorem 2.** *Suppose Assumptions 1 and 2 hold, the following holds with probability at least $1 - \delta$:*

$$SubOptR(\pi_{\phi^*, \psi^*, \lambda^*(\phi^*, \psi^*)}) \leq \mathcal{O}(\sqrt{\frac{1}{n} \ln(\frac{N(\mathcal{F}, \|\cdot\|_\infty, \frac{1}{n})}{\delta})}),$$

*and*

$$SubOptC(\pi_{\phi^*, \psi^*, \lambda^*(\phi^*, \psi^*)}) \leq \mathcal{O}(\sqrt{\frac{1}{n} \ln(\frac{N(\mathcal{F}, \|\cdot\|_\infty, \frac{1}{n})}{\delta})}).$$

Theorem 2 demonstrates that the policy $\pi_{\phi^*, \psi^*, \lambda^*(\phi^*, \psi^*)}$ is learned efficiently.

# 7 EXPERIMENT

This section presents the experimental results of Algorithm 1 in the MuJoCo environment (Towers et al., 2024), a widely used physics engine and simulation platform for modeling and controlling complex robotic systems in reinforcement learning and robotics research. We evaluate CB-RLHF

on four benchmark tasks, Walker2D, Half Cheetah, Hopper, and Swimmer. In each environment, the default reward function is adopted as the ground-truth reward, and a corresponding ground-truth cost function is designed. Detailed descriptions of these environments and the construction of the cost functions are provided in the Appendix A.10.

We compare the proposed algorithm with three baseline methods: PEBBLE (Lee et al., 2021), Safe RLHF (Dai et al., 2023), and PARL (Chakraborty et al., 2023). PEBBLE, a standard RLHF algorithm, suffers from both the constraint inference and misalignment limitations. Safe RLHF addresses the constraint inference limitation, while PARL mitigates the misalignment issue. In contrast, CB-RLHF simultaneously tackles both. During training, human feedback is synthetically generated based on the ground-truth reward and cost functions. For each trajectory pair, the trajectory with the higher cumulative reward under the ground-truth reward function is preferred in terms of optimality, while the trajectory with the lower cumulative cost under the ground-truth cost function is preferred in terms of constraint satisfaction. A trajectory is considered to violate the constraint if its cumulative cost exceeds zero.

Table 1: Cumulative return and constraint violation rate at the final environment step.

| Environment | Algorithm | Cumulative return | Constraint violation rate |
|---|---|---|---|
| Walker2d | PEBBLE | $750.46 \pm 101.82$ | $0.99 \pm 0.01$ |
| | Safe RLHF | $1069.51 \pm 138.74$ | $0.98 \pm 0.01$ |
| | PARL | $1086.01 \pm 210.05$ | $0.91 \pm 0.07$ |
| | CB-RLHF | $1157.41 \pm 183.75$ | $0.87 \pm 0.07$ |
| HalfCheetah | PEBBLE | $935.89 \pm 90.71$ | $0.06 \pm 0.05$ |
| | Safe RLHF | $100.92 \pm 163.55$ | $0.13 \pm 0.10$ |
| | PARL | $1933.58 \pm 429.37$ | $0.29 \pm 0.07$ |
| | CB-RLHF | $1849.94 \pm 36.59$ | $0.13 \pm 0.08$ |
| Hopper | PEBBLE | $164.12 \pm 117.90$ | $0.69 \pm 0.30$ |
| | Safe RLHF | $329.08 \pm 17.67$ | $0.52 \pm 0.23$ |
| | PARL | $303.94 \pm 22.30$ | $0.60 \pm 0.37$ |
| | CB-RLHF | $292.45 \pm 10.77$ | $0.33 \pm 0.21$ |
| Swimmer | PEBBLE | $-8.45 \pm 14.17$ | $0.02 \pm 0.01$ |
| | Safe RLHF | $38.45 \pm 6.87$ | $0.17 \pm 0.07$ |
| | PARL | $70.22 \pm 14.37$ | $0.29 \pm 0.14$ |
| | CB-RLHF | $73.19 \pm 10.65$ | $0.02 \pm 0.02$ |

For each cumulative return calculation, we evaluate the policy learned at the current environment step by sampling 100 trajectories. We then compute the average cumulative reward under the ground-truth reward function and the average cumulative cost under the ground-truth cost function, defining the cumulative return as the cumulative reward minus the cumulative cost. Similarly, each learned policy is assessed over 100 independent trials to estimate the constraint violation rate, defined as the fraction of trials in which the agent violates the constraint.

From Table 1, excluding the unsuccessful learning cases (e.g., Safe RLHF in Half Cheetah), we observe that at the final environment step, PARL and CB-RLHF achieve higher cumulative returns than Safe RLHF and PEBBLE, confirming that misalignment can lead to suboptimal policies. At the same time, CB-RLHF and Safe RLHF achieve lower constraint violation rates, indicating that learning a cost function enables the agent to better satisfy constraints. Overall, these results demonstrate that CB-RLHF simultaneously addresses both the misalignment and constraint inference limitations.

## 8  CONCLUSION

We develop a constrained RLHF framework that enables an agent to jointly learn a reward function, a cost function, and a policy from human feedback. This framework simultaneously addresses the misalignment and constraint inference limitations. Building on this formulation, we introduced the CB-RLHF algorithm and established theoretical guarantees, including its convergence rate and the performance gap between the optimal and learned policies. Experiments on MuJoCo environments further validate the effectiveness of CB-RLHF, demonstrating its ability to achieve strong performance while ensuring constraint satisfaction.

## 9 ETHICS STATEMENT

This work does not involve human subjects or sensitive personal data. All experiments are conducted on publicly available simulation benchmarks.

## 10 REPRODUCIBILITY STATEMENT

The details of the experimental setup are provided in Appendix A.10, and the source code is included in the supplementary material to facilitate reproducibility.

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

# A  APPENDIX

## A.1  NOTION AND NOTATIONS

We define some equations used in the appendix as follows:

- The casual entropy from state $s$ and action $a$ :
  $H(\pi, s, a) = -\ln(\pi(a|s)) + \gamma \int_{s' \in S} P(s'|s, a) H(\pi, s') ds'$.
- the casual entropy from state $s$:
  $H(\pi, s) = \int_{a \in A} \pi(a|s) H(\pi, s, a) da$.
- The cumulative reward from state $s$ and action $a$:
  $J_r(\pi, s, a) = r(s, a) + \gamma \int_{s' \in S} P(s'|s, a) J_r(\pi, s') ds'$.
- the cumulative reward from state $s$:
  $J_r(\pi, s) = \int_{a \in A} \pi(a|s) J_r(\pi, s, a) da$.
- the cumulative cost from state $s$ and action $a$:
  $J_c(\pi, s, a) = c(s, a) + \gamma \int_{s' \in S} P(s'|s, a) J_c(\pi, s') ds'$
- the cumulative cost from state $s$:
  $J_c(\pi, s) = \int_{a \in A} \pi(a|s) J_c(\pi, s, a) da$.

During the proofs, we use some symbols to label the methods and theorems used for deriving the current equation. The symbols and what they represent are as follows:

- The symbol $(i)$ represents chain rule.
- The symbol $(ii)$ represents the linearity of expectation.
- The symbol $(iii)$ represents the close form of geometric series.
- The symbol $(iv)$ represents the triangle inequality.
- The symbol $(v)$ represents the hölder's inequality.
- The symbol $(vi)$ represents Taylor's theorem.
- The symbol $(vii)$ means the usage of other equations in this paper.
- The symbol $(viii)$ means keeping expansion.

Based on the idea of soft Q learning (Haarnoja et al., 2017), the formula of the constrained soft Bellman policy for continuous state-action space is as follows:

$$\pi_{\phi,\psi,\lambda}(a|s) = \frac{\exp(Q_{\phi,\psi,\lambda}^{soft}(s,a))}{\exp(V_{\phi,\psi,\lambda}^{soft}(s))}, \tag{11}$$

$$Q_{\phi,\psi,\lambda}^{soft}(s,a) = r_\phi(s,a) - \lambda c_\psi(s,a) + \gamma \int_{s' \in S} P(s'|s,a) V_{\phi,\psi,\lambda}^{soft}(s') ds', \tag{12}$$

$$V_{\phi,\psi,\lambda}^{soft}(s) = \ln \int_{a \in A} \exp(Q_{\phi,\psi,\lambda}^{soft}(s,a)) da. \tag{13}$$

From the expression of $\pi_{\phi,\psi,\lambda}$, we can get $\nabla_\lambda \ln(\pi_{\phi,\psi,\lambda}) = \nabla_\lambda Q_{\phi,\psi,\lambda}^{soft}(s,a) - \nabla_\lambda V_{\phi,\psi,\lambda}^{soft}(s)$, therefore, we can calculate $\nabla_\psi Q_{\phi,\psi,\lambda}^{soft}(s,a)$ and $\nabla_\psi V_{\phi,\psi,\lambda}^{soft}(s)$ separately.

Based on the equation of $V^{soft}(s)$, the gradient $\nabla_\psi V^{soft}(s)$ could be calculated as follows:

$$
\begin{aligned}
&\nabla_\lambda V_{\phi,\psi,\lambda}^{soft}(s), \\
&\overset{(i)}{=} \frac{\int_{a \in A} \nabla_\lambda \exp(Q_{\phi,\psi,\lambda}^{soft}(s,a)) da}{\int_{a \in A} \exp(Q_{\phi,\psi,\lambda}^{soft}(s,a)) da}, \\
&\overset{(vii)}{=} \frac{\int_{a \in A} \exp(Q_{\phi,\psi,\lambda}^{soft}(s,a)) \nabla_\lambda Q_{\phi,\psi,\lambda}^{soft}(s,a) da}{\exp(V_{\phi,\psi,\lambda}^{soft}(s))}, \\
&\overset{(vii)}{=} \int_{a \in A} \pi_{\phi,\psi,\lambda}(a|s)(-c_\psi(s,a) + \gamma \int_{s' \in S} P(s'|s,a) \nabla_\lambda V_{\phi,\psi,\lambda}^{soft}(s') ds') da, \\
&\overset{(viii)}{=} \int_{a \in A} \pi_{\phi,\psi,\lambda}(a|s)(-c_\psi(s,a) + \gamma \int_{s' \in S} P(s'|s,a) \int_{a' \in A} \pi_{\phi,\psi,\lambda}(a'|s')[-c_\psi(s',a') \\
&\quad + \gamma \int_{s'' \in S} P(s''|s',a') \nabla_\lambda V_{\phi,\psi,\lambda}^{soft}(s'') ds''] da' ds') da, \\
&= E^{\pi_{\phi,\psi,\lambda}} \Big[ \sum_{h=0}^{H-1} -\gamma^h c_\psi(s_h,a_h)|s_0 = s \Big] \\
&= -J_{c_\psi}(\pi^{\phi,\psi,\lambda}, s),
\end{aligned} \tag{14}
$$

where the first $(vii)$ uses the expression in equation (13) and (11), and the second $(iii)$ uses the expression in equation (12).

Similarly, we can get $\nabla_\lambda Q_{\phi,\psi,\lambda}^{soft}(s,a)$ from $Q_{\phi,\psi,\lambda}^{soft}(s,a)$.

$$
\begin{aligned}
&\nabla_\lambda Q_{\phi,\psi,\lambda}^{soft}(s,a), \\
&\overset{(vii)}{=} -c_\psi(s,a) + \gamma \int_{s' \in S} P(s'|s,a) \nabla_\lambda V_{\phi,\psi,\lambda}^{soft}(s') ds') da, \\
&\overset{(viii)}{=} E^{\pi_{\phi,\psi,\lambda}} \Big[ \sum_{h=0}^{H-1} -\gamma^h c_\psi(s_h,a_h)|s_0 = s, a_0 = a \Big] \\
&= -J_{c_\psi}(\pi^{\phi,\psi,\lambda}, s, a),
\end{aligned} \tag{15}
$$

where $(vii)$ uses the expression in equation (12).

By summing the results of equation (14) and (15), we can get $\nabla_\psi \ln(\pi^{\phi,\psi})$ as follows:

$$
\begin{aligned}
\nabla_\lambda \ln(\pi_{\phi,\psi,\lambda}(a_1,a_2|s)) &= \nabla_\lambda Q_{\phi,\psi,\lambda}^{soft}(s,a_1,a_2) - \nabla_\lambda V_{\phi,\psi,\lambda}^{soft}(s), \\
&= J_{c_\psi}(\pi_{\phi,\psi,\lambda}, s) - J_{c_\psi}(\pi_{\phi,\psi,\lambda}, s, a).
\end{aligned} \tag{16}
$$

## A.2 IMPORTANT GRADIENTS

In this section, the necessary gradients for future proofs are calculated. First, the gradients respect to $\lambda$ are shown as follows:

$$
\nabla_\lambda H(\pi_{\phi,\psi,\lambda})
$$

$$
= \int_{s_0 \in S} P_0(s_0) \int_{a_0 \in A} \nabla_\lambda \pi_{\phi,\psi,\lambda}(a_0|s_0) H(\pi_{\phi,\psi,\lambda}, s_0, a_0) + \pi_{\phi,\psi,\lambda}(a_0|s_0) \nabla_\lambda H(\pi_{\phi,\psi,\lambda}, s_0, a_0) da_0 ds_0
$$

$$
= \int_{s_0 \in S} P_0(s_0) \int_{a_0 \in A} \nabla_\lambda \pi_{\phi,\psi,\lambda}(a_0|s_0) H(\pi_{\phi,\psi,\lambda}, s_0, a_0) + \pi_{\phi,\psi,\lambda}(a_0|s_0) \nabla_\lambda (-\ln(\pi_{\phi,\psi,\lambda})(a_0|s_0)
$$

$$
+ \gamma \int_{s_1 \in S} P(s_1|s_0, a_0) H(\pi_{\phi,\psi,\lambda}, s_1) ds_1) da_0 ds_0
$$

$$
= \int_{s_0 \in S} P_0(s_0) \int_{a_0 \in A} \pi_{\phi,\psi,\lambda}(a_0|s_0) \nabla_\lambda \ln(\pi_{\phi,\psi,\lambda}(a_0|s_0)) H(\pi_{\phi,\psi,\lambda}, s_0, a_0)
$$

$$
- \pi_{\phi,\psi,\lambda}(a_0|s_0) \nabla_\lambda \ln(\pi_{\phi,\psi,\lambda})(a_0|s_0) + \gamma \int_{s_1 \in S} P(s_1|s_0, a_0) \int_{a_1 \in A} \nabla_\lambda \pi_{\phi,\psi,\lambda}(a_1|s_1) H(\pi_{\phi,\psi,\lambda}, s_1, a_1)
$$

$$
+ \pi_{\phi,\psi,\lambda}(a_1|s_1) \nabla_\lambda H(\pi_{\phi,\psi,\lambda}, s_1, a_1) da_1 ds_1 da_0 ds_0
$$

$$
= \int_{s_0 \in S} P_0(s_0) \int_{a_0 \in A} \pi_{\phi,\psi,\lambda}(a_0|s_0) \nabla_\lambda \ln(\pi_{\phi,\psi,\lambda}(a_0|s_0)) H(\pi_{\phi,\psi,\lambda}, s_0, a_0)
$$

$$
- \pi_{\phi,\psi,\lambda}(a_0|s_0) \nabla_\lambda \ln(\pi_{\phi,\psi,\lambda})(a_0|s_0) + \gamma \int_{s_1 \in S} P(s_1|s_0, a_0) \int_{a_1 \in A} \pi_{\phi,\psi,\lambda}(a_1|s_1) \nabla_\lambda \ln(\pi_{\phi,\psi,\lambda}(a_1|s_1))
$$

$$
H(\pi_{\phi,\psi,\lambda}, s_1, a_1) + \pi_{\phi,\psi,\lambda}(a_1|s_1) \nabla_\lambda (-\ln(\pi_{\phi,\psi,\lambda}(a_1|s_1))
$$

$$
+ \gamma \int_{s_2 \in S} P(s_2|s_1, a_1) H(\pi_{\phi,\psi,\lambda}, s_2) ds_2) da_1 ds_1 da_0 ds_0
$$

$$
= \int_{s \in S} f(\pi_{\phi,\psi,\lambda}, s) \int_{a \in A} \pi_{\phi,\psi,\lambda}(a|s) \nabla_\lambda \ln(\pi_{\phi,\psi,\lambda}(a|s))(H(\pi_{\phi,\psi,\lambda}, s, a) - 1) da ds
$$

$$
\nabla_\lambda J_{r_\phi}(\pi_{\phi,\psi,\lambda})
$$

$$
= \int_{s_0 \in S} P_0(s_0) \int_{a_0 \in A} \nabla_\lambda \pi_{\phi,\psi,\lambda}(a_0|s_0) J_{r_\phi}(\pi_{\phi,\psi,\lambda}, s_0, a_0) + \pi_{\phi,\psi,\lambda}(a_0|s_0) \nabla_\lambda J_{r_\phi}(\pi_{\phi,\psi,\lambda}, s_0, a_0) da_0 ds_0
$$

$$
= \int_{s_0 \in S} P_0(s_0) \int_{a_0 \in A} \nabla_\lambda \pi_{\phi,\psi,\lambda}(a_0|s_0) J_{r_\phi}(\pi_{\phi,\psi,\lambda}, s_0, a_0) + \pi_{\phi,\psi,\lambda}(a_0|s_0) \nabla_\lambda r_\psi(s_0, a_0)
$$

$$
+ \gamma \int_{s_1 \in S} P(s_1|s_0, a_0) J_{r_\phi}(\pi_{\phi,\psi,\lambda}, s_1) ds_1) da_0 ds_0
$$

$$
= \int_{s_0 \in S} P_0(s_0) \int_{a_0 \in A} \nabla_\lambda \pi_{\phi,\psi,\lambda}(a_0|s_0) J_{r_\phi}(\pi_{\phi,\psi,\lambda}, s_0, a_0) + \pi_{\phi,\psi,\lambda}(a_0|s_0) \gamma \int_{s_1 \in S} P(s_1|s_0, a_0)
$$

$$
\int_{a_1 \in A} \nabla_\lambda \pi_{\phi,\psi,\lambda}(a_1|s_1) J_{r_\phi}(\pi_{\phi,\psi,\lambda}, s_1, a_1) + \pi_{\phi,\psi,\lambda}(a_1|s_1) \nabla_\lambda J_{r_\phi}(\pi_{\phi,\psi,\lambda}, s_1, a_1) da_1 ds_1 da_0 ds_0
$$

$$
= \int_{s \in S} f(\pi_{\phi,\psi,\lambda}, s) \int_{a \in A} \pi_{\phi,\psi,\lambda}(a|s) \nabla_\lambda \ln(\pi_{\phi,\psi,\lambda}(a|s)) J_{r_\phi}(\pi_{\phi,\psi,\lambda}, s, a) da ds
$$

Through the same process, we can get $\nabla_\lambda J_{c_\psi}(\pi_{\phi,\psi,\lambda})$ as follows:

$$
\nabla_\lambda J_{c_\psi}(\pi_{\phi,\psi,\lambda}) = \int_{s \in S} f(\pi_{\phi,\psi,\lambda}, s) \int_{a \in A} \pi_{\phi,\psi,\lambda}(a|s) \nabla_\lambda \ln(\pi_{\phi,\psi,\lambda}(a|s)) J_{c_\psi}(\pi_{\phi,\psi,\lambda}, s, a) da ds
$$

The gradients respect to $\psi$ are shown as follows:

$$\nabla_\psi H(\pi_{\phi,\psi}, s, a)$$

$$= -\nabla_\psi \ln(\pi_{\phi,\psi}(a|s)) + \gamma \int_{s' \in S} P(s'|s,a) \nabla_\psi H(\pi_{\phi,\psi}, s') ds',$$

$$\overset{(i)}{=} -\nabla_\psi \ln(\pi_{\phi,\psi}(a|s)) + \gamma \int_{s' \in S} P(s'|s,a) \int_{a' \in A} [\pi_{\phi,\psi}(a'|s') \nabla_\psi H(\pi_{\phi,\psi} s', a')$$

$$+ \nabla_\psi \pi_{\phi,\psi}(a'|s') H(\pi_{\phi,\psi}, s', a')] da' ds',$$

$$\overset{(i)}{=} -\nabla_\psi \ln(\pi_{\phi,\psi}(a|s)) + \gamma \int_{s' \in S} P(s'|s,a) \int_{a' \in A} \pi_{\phi,\psi}(a'|s')[\nabla_\psi H(\pi_{\phi,\psi}, s', a')$$

$$+ \nabla_\psi \ln(\pi_{\phi,\psi}(a'|s')) H(\pi_{\phi,\psi}, s', a')] da' ds',$$

$$\overset{(viii)}{=} -\nabla_\psi \ln(\pi_{\phi,\psi}(a|s)) + \gamma \int_{s' \in S} P(s'|s,a) \int_{a' \in A} \pi_{\phi,\psi}(a'|s')[-\nabla_\psi \ln(\pi_{\phi,\psi}(a'|s'))$$

$$+ \gamma \int_{s'' \in S} P(s''|s', a') \nabla_\psi H(\pi_{\phi,\psi}, s'') ds'' + \nabla_\psi \ln(\pi_{\phi,\psi}(a'|s')) H(\pi_{\phi,\psi}, s', a')] da' ds',$$

$$\overset{(viii)}{=} -\nabla_\psi \ln(\pi_{\phi,\psi}(a|s)) + E^{\pi_{\phi,\psi}} \left[ \sum_{h=1}^{H-1} \gamma^h \nabla_\psi \ln(\pi_{\phi,\psi}(a_h|s_h))(H(\pi_{\phi,\psi}, s_h, a_h) - 1)|s_0 = s, a_0 = a \right]$$

$$\nabla_\psi H(\pi_{\phi,\psi}, s) = \int_{a \in A} \pi(a|s) H(\pi_{\phi,\psi}, s, a) da$$

$$= E^{\pi_{\phi,\psi}} \left[ \sum_{h=0}^{H-1} \gamma^h \nabla_\psi \ln(\pi_{\phi,\psi}(a_h|s_h))(H(\pi_{\phi,\psi}, s_h, a_h) - 1)|s_0 = s \right]$$

$$\nabla_\psi J_{r_\phi}(\pi_{\phi,\psi}, s, a)$$

$$= \gamma \int_{s' \in S} P(s'|s,a) \nabla_\psi J_{r_\phi}(\pi_{\phi,\psi}, s') ds',$$

$$\overset{(i)}{=} +\gamma \int_{s' \in S} P(s'|s,a) \int_{a' \in A} [\pi_{\phi,\psi}(a'|s') \nabla_\psi J_{r_\phi}(\pi_{\phi,\psi}, s', a')$$

$$+ \nabla_\psi \pi_{\phi,\psi}(a'|s') J_{r_\phi}(\pi_{\phi,\psi}, s', a')] da' ds',$$

$$\overset{(i)}{=} \gamma \int_{s' \in S} P(s'|s,a) \int_{a' \in A} \pi_{\phi,\psi}(a'|s')[\nabla_\psi J_{r_\phi}(\pi_{\phi,\psi}, s', a')$$

$$+ \nabla_\psi \ln(\pi_{\phi,\psi}(a'|s')) J_{r_\phi}(\pi_{\phi,\psi}, s', a')] da' ds',$$

$$\overset{(viii)}{=} \gamma \int_{s' \in S} P(s'|s,a) \int_{a' \in A} \pi_{\phi,\psi}(a'|s')[-\nabla_\psi \ln(\pi_{\phi,\psi}(a'|s'))$$

$$+ \gamma \int_{s'' \in S} P(s''|s', a') \nabla_\psi J_{r_\phi}(\pi_{\phi,\psi}, s'') ds'' + \nabla_\psi \ln(\pi_{\phi,\psi}(a'|s')) J_{r_\phi}(\pi_{\phi,\psi}, s', a')] da' ds',$$

$$\overset{(viii)}{=} E^{\pi_{\phi,\psi}} \left[ \sum_{h=1}^{H-1} \gamma^h \nabla_\psi \ln(\pi_{\phi,\psi}(a_h|s_h)) J_{r_\phi}(\pi_{\phi,\psi}, s_h, a_h)|s_0 = s, a_0 = a \right]$$

$$\nabla_\psi J_{r_\phi}(\pi_{\phi,\psi}, s) = \int_{a \in A} \pi_{\phi,\psi}(a|s) J_{r_\phi}(\pi_{\phi,\psi}, s, a) da$$

$$= E^{\pi_{\phi,\psi}} \left[ \sum_{h=0}^{H-1} \gamma^h \nabla_\psi \ln(\pi_{\phi,\psi}(a_h|s_h)) J_{r_\phi}(\pi_{\phi,\psi}, s_h, a_h)|s_0 = s \right]$$

$$\nabla_\psi J_{c_\psi}(\pi_{\phi,\psi}, s, a)$$

$$= \nabla_\psi c_\psi(s, a) + \gamma \int_{s' \in S} P(s'|s, a) \nabla_\psi J_{c_\psi}(\pi_{\phi,\psi}, s') ds',$$

$$\overset{(i)}{=} \nabla_\psi c_\psi(s, a) + \gamma \int_{s' \in S} P(s'|s, a) \int_{a' \in A} [\pi_{\phi,\psi}(a'|s') \nabla_\psi J_{c_\psi}(\pi_{\phi,\psi} s', a')$$

$$+ \nabla_\psi \pi_{\phi,\psi}(a'|s') J_{c_\psi}(\pi_{\phi,\psi}, s', a')] da' ds',$$

$$\overset{(i)}{=} \nabla_\psi c_\psi(s, a) + \gamma \int_{s' \in S} P(s'|s, a) \int_{a' \in A} \pi_{\phi,\psi}(a'|s') [\nabla_\psi J_{c_\psi}(\pi_{\phi,\psi}, s', a')$$

$$+ \nabla_\psi \ln(\pi_{\phi,\psi}(a'|s')) J_{c_\psi}(\pi_{\phi,\psi}, s', a')] da' ds',$$

$$\overset{(viii)}{=} \nabla_\psi c_\psi(s, a) + \gamma \int_{s' \in S} P(s'|s, a) \int_{a' \in A} \pi_{\phi,\psi}(a'|s') [\nabla_\psi c_\psi(a'|s')$$

$$+ \gamma \int_{s'' \in S} P(s''|s', a') \nabla_\psi J_{c_\psi}(\pi_{\phi,\psi}, s'') ds'' + \nabla_\psi \ln(\pi_{\phi,\psi}(a'|s')) J_{c_\psi}(\pi_{\phi,\psi}, s', a')] da' ds',$$

$$\overset{(viii)}{=} \nabla_\psi c_\psi(s, a) + E^{\pi_{\phi,\psi}} [\sum_{h=1}^{H-1} \gamma^h (\nabla_\psi c_\psi(s, a) + \nabla_\psi \ln(\pi_{\phi,\psi}(a_h|s_h)) J_{c_\psi}(\pi_{\phi,\psi}, s_h, a_h)) | s_0 = s, a_0 = a]$$

$$\nabla_\psi J_{c_\psi}(\pi_{\phi,\psi}, s) = \int_{a \in A} \pi_{\phi,\psi}(a|s) J_{c_\psi}(\pi_{\phi,\psi}, s, a) da$$

$$= E^{\pi_{\phi,\psi}} [\sum_{h=0}^{H-1} \gamma^h (\nabla_\psi c_\psi(s, a) + \nabla_\psi \ln(\pi_{\phi,\psi}(a_h|s_h)) J_{c_\psi}(\pi_{\phi,\psi}, s_h, a_h)) | s_0 = s]$$

### A.3   PROOF OF LEMMA 1

The Lagranian function of the constrained RL problem is $L(\pi; \phi, \psi, \lambda) \triangleq E^\pi [\sum_{h=0}^{H-1} \gamma^h (r_\phi(s_h, a_h) + \ln(\pi(a_h|s_h)) - \lambda c_\psi(s_h, a_h))$. From (Haarnoja et al., 2017), the optimal solution for $\arg\max\limits_{\pi \in \Pi} L(\pi; \phi, \psi, \lambda)$ exists, and it is the continuous constrained soft Bellman policy. We assume the policy $\pi$ is time-dependent but stationary, the partial derivative according to $\pi^h$ is as follows:

$$\frac{\partial L(\pi; \phi, \psi, \lambda)}{\partial \pi^h(a|s)}$$

$$\overset{(i)}{=} P(s_h = s) \{ \gamma^h (\ln(\pi^h(a|s)) + 1) + E^\pi [\sum_{i=h+1}^{H-1} \gamma^h \ln(\pi^i(a_h|s_h)) | s_h = s, a_h = a]$$

$$+ \gamma^h (r_\phi(s_h, a_h) - \lambda c_\psi(s_h, a_h)) + E^\pi [\sum_{i=h+1}^{H-1} \gamma^h (r_\phi(s_h, a_h) - \lambda c_\psi(s_h, a_h)) | s_h = s, a_h = a] \},$$

$$= P(s_h = s) \gamma^h (1 - H(\pi, s, a) + J_r(\pi, s, a) - \lambda J_c(\pi, s, a)),$$

where $P(s_h = s)$ represents the probability (density) of reaching state $s$ at time $h$. Since $\pi(a|s) > 0$ for all $a \in A$, $s \in S$ according to equation (11), the Lagranian function $L(\pi; \phi, \psi, \lambda)$ reaches the maximum when $\frac{\partial L(\pi; \phi, \psi, \lambda)}{\partial \pi^h(a|s)} = 0$ with $P(s_h = s) \neq 0$ for at least one time. We can get $1 - H(\pi_{\phi,\psi,\lambda}, s, a) - J_{r_\phi}(\pi_{\phi,\psi,\lambda}, s, a) + \lambda J_{c_\psi}(\pi_{\phi,\psi,\lambda}, s, a) = 0$. The dual function $\min_{\lambda \geq 0} G(\lambda; \phi, \psi)$. The derivative of $G(\lambda; \phi, \psi)$ according to $\lambda$ is as follows:

$$\nabla_\lambda G(\lambda; \phi, \psi)$$

$$= \nabla_\lambda H(\pi_{\phi,\psi,\lambda})) + \nabla_\lambda J_{r_\phi}(\pi_{\phi,\psi,\lambda}) - \lambda \nabla_\lambda J_{c_\psi}(\pi_{\phi,\psi,\lambda}) - J_{c_\psi}(\pi_{\phi,\psi,\lambda})$$

$$= \int_{s \in S} f(\pi_{\phi,\psi,\lambda}, s) \int_{a \in A} \pi_{\phi,\psi,\lambda}(a|s) \nabla_\lambda \ln(\pi_{\phi,\psi,\lambda}(a|s))(1 - H(\pi_{\phi,\psi,\lambda}, s, a) + J_{r_\phi}(\pi_{\phi,\psi,\lambda}, s, a)$$

$$- \lambda J_{c_\psi}(\pi_{\phi,\psi,\lambda}, s, a)) da ds - J_{c_\psi}(\pi_{\phi,\psi,\lambda})$$

When $G(\lambda; \phi, \psi)$ reach the minimum, the gradient $\nabla_\lambda G(\lambda; \phi, \psi)$ should be 0 which means $J_{c_\psi}(\pi_{\phi,\psi,\lambda^*(\phi,\psi)}) = 0$. We can see that $\pi_{\phi,\psi,\lambda^*(\phi,\psi)}$ is a feasible solution for the constrained RL problem. However, $\lambda^*$ should be non-negative. If the calculated $\lambda^*$ is non-negative, we can get $H(\pi_{\phi,\psi,\lambda^*(\phi,\psi)}) + J_{r_\phi}(\pi_{\phi,\psi,\lambda^*(\phi,\psi)}) \leq p^* \leq d^* = G(\lambda^*(\phi,\psi); \phi, \psi) = H(\pi_{\phi,\psi,\lambda^*(\phi,\psi)}) + J_{r_\phi}(\pi_{\phi,\psi,\lambda^*(\phi,\psi)})$ where $p^*$ is the maximum value for the primal problem and $d^*$ is the minimal value of $\min_{\lambda \geq 0} G(\lambda; \phi, \psi)$. If the calculated $\lambda^*$ is negative, the final result for $\lambda^*(\phi, \psi)$ is 0. Therefore, we can get $H(\pi_{\phi,\psi,0}) + J_{r_\phi}(\pi_{\phi,\psi,0}) = G(0; \phi, \psi)$ which means $\pi_{\phi,\psi,0}$ is still a feasible policy for the primal problem. Overall, the strong duality holds for the prime problem and the dual problem.

### A.4 HYPERGRADIENT CALCULATION

In order to simplify the computation, we use $f(\tau_0, \tau_1) = (1 - y_r) \log \sigma(J_{r_\phi}(\tau_0) - J_{r_\phi}(\tau_1)) + y_r \log \sigma(J_{r_\phi}(\tau_1) - J_{r_\phi}(\tau_0)) + (1 - y_c) \log \sigma(J_{c_\psi}(\tau_0) - J_{c_\psi}(\tau_1)) + y_c \log \sigma(J_{c_\psi}(\tau_1) - J_{c_\psi}(\tau_0)) + \log \sigma(s_0 J_{c_\psi}(\tau_0)) + \log \sigma(s_1 J_{c_\psi}(\tau_1))$ for the following calculation.

Define $\mu_\phi(\pi) = E^\pi[\nabla_\phi r_\phi]$ and $\mu_\psi(\pi) = E^\pi[\nabla_\psi c_\psi]$

The hypergradient corresponding to $\phi$ is calculated as follows:

$$\nabla_\phi F(\phi, \psi, \lambda^*(\phi, \psi))$$

$$= \nabla_\phi \sum_{(\tau_0, \tau_1)} f(\tau_0, \tau_1) D(\tau; \lambda^*(\phi, \psi))$$

$$= \sum_{(\tau_0, \tau_1)} \nabla_\phi f(\tau_0, \tau_1) D(\tau; \lambda^*(\phi, \psi)) + f(\tau_0, \tau_1) \nabla_\phi D(\tau; \lambda^*(\phi, \psi))$$

$$= \sum_{(\tau_0, \tau_1)} \nabla_\phi f(\tau_0, \tau_1) D(\tau; \lambda^*(\phi, \psi)) + f(\tau_0, \tau_1) \nabla_\phi \ln(D(\tau; \lambda^*(\phi, \psi))) D(\tau; \lambda^*(\phi, \psi))$$

$$= E_{D(\tau; \lambda^*(\phi,\psi))}[\nabla_\phi f(\tau_0, \tau_1) + f(\tau_0, \tau_1) \nabla_\phi (\ln(h_l(y_r, y_c, s_0, s_1 | \tau_0, \tau_1)) + \ln(\rho(\tau_0; \lambda^*(\phi, \psi)))$$

$$+ \ln(\rho(\tau_1; \lambda^*(\phi, \psi))))]$$

$$= E_{D(\tau; \lambda^*(\phi,\psi))}[\nabla_\phi f(\tau_0, \tau_1) + f(\tau_0, \tau_1)(\nabla_\phi \ln(\rho(\tau_0; \lambda^*(\phi, \psi))) + \nabla_\phi \ln(\rho(\tau_1; \lambda^*(\phi, \psi))))]$$

$$= E_{D(\tau; \lambda^*(\phi,\psi))}[\nabla_\phi f(\tau_0, \tau_1) + f(\tau_0, \tau_1)(\sum_{h=0}^{H-1} \nabla_\phi \ln(\pi_{\phi,\psi,\lambda^*(\phi,\psi)}(a_h^0 | s_h^0))$$

$$+ \sum_{h=0}^{H-1} \nabla_\phi \ln(\pi_{\phi,\psi,\lambda^*(\phi,\psi)}(a_h^1 | s_h^1)))],$$

where $a_h^0, s_h^0 \in \tau_0$ and $a_h^1, s_h^1 \in \tau_1$. Analogously, we can get

$$\nabla_\psi F(\phi, \psi, \lambda^*(\phi, \psi))$$

$$= E_{D(\tau; \lambda^*(\phi,\psi))}[\nabla_\psi f(\tau_0, \tau_1) + f(\tau_0, \tau_1)(\sum_{h=0}^{H-1} \nabla_\psi \ln(\pi_{\phi,\psi,\lambda^*(\phi,\psi)}(a_h^0 | s_h^0))$$

$$+ \sum_{h=0}^{H-1} \nabla_\psi \ln(\pi_{\phi,\psi,\lambda^*(\phi,\psi)}(a_h^1 | s_h^1)))],$$

and

$$\nabla_\lambda F(\phi, \psi, \lambda^*(\phi, \psi))$$

$$= E_{D(\tau; \lambda^*(\phi,\psi))}[f(\tau_0, \tau_1)(\sum_{h=0}^{H-1} \nabla_\lambda \ln(\pi_{\phi,\psi,\lambda^*(\phi,\psi)}(a_h^0|s_h^0))$$

$$+ \sum_{h=0}^{H-1} \nabla_\lambda \ln(\pi_{\phi,\psi,\lambda^*(\phi,\psi)}(a_h^1|s_h^1)))]$$

Next the terms in $\nabla_\phi F(\phi, \psi, \lambda^*(\phi, \psi))$, $\nabla_\psi F(\phi, \psi, \lambda^*(\phi, \psi))$, and $\nabla_\lambda F(\phi, \psi, \lambda^*(\phi, \psi))$ are calculated as follows:

$$\nabla_\phi \ln(\sigma(J_{r_\phi}(\tau_0) - J_{r_\phi}(\tau_1)))$$

$$= \nabla_\phi \ln(\frac{\exp(J_{r_\phi}(\tau_0))}{\exp(J_{r_\phi}(\tau_0)) + \exp(J_{r_\phi}(\tau_1))})$$

$$= \nabla_\phi J_{r_\phi}(\tau_0) - \nabla_\phi \ln(\exp(J_{r_\phi}(\tau_0)) + \exp(J_{r_\phi}(\tau_1)))$$

$$= \nabla_\phi J_{r_\phi}(\tau_0) + - \frac{\nabla_\phi J_{r_\phi}(\tau_0) \exp(J_{r_\phi}(\tau_0)) + \nabla_\phi J_{r_\phi}(\tau_1) \exp(J_{r_\phi}(\tau_1))}{\exp(J_{r_\phi}(\tau_0)) + \exp(J_{r_\phi}(\tau_1))}$$

Analogously, we can get

$$\nabla_\phi \ln(\sigma(J_{r_\phi}(\tau_1) - J_{r_\phi}(\tau_0))) = \nabla_\phi J_{r_\phi}(\tau_1) - \frac{\nabla_\phi J_{r_\phi}(\tau_0) \exp(J_{r_\phi}(\tau_0)) + \nabla_\phi J_{r_\phi}(\tau_1) \exp(J_{r_\phi}(\tau_1))}{\exp(J_{r_\phi}(\tau_0)) + \exp(J_{r_\phi}(\tau_1))}$$

$$\nabla_\phi \ln(\sigma(J_{c_\psi}(\tau_0) - J_{c_\psi}(\tau_1))) = 0$$

$$\nabla_\phi \ln(\sigma(J_{c_\psi}(\tau_1) - J_{c_\psi}(\tau_0))) = 0$$

$$\nabla_\psi \ln(\sigma(J_{c_\psi}(\tau_0) - J_{c_\psi}(\tau_1))) = \nabla_\psi J_{c_\psi}(\tau_0) - \frac{\nabla_\psi J_{c_\psi}(\tau_0) \exp(J_{c_\psi}(\tau_0)) + \nabla_\psi J_{c_\psi}(\tau_1) \exp(J_{c_\psi}(\tau_1))}{\exp(J_{c_\psi}(\tau_0)) + \exp(J_{c_\psi}(\tau_1))}$$

$$\nabla_\psi \ln(\sigma(J_{c_\psi}(\tau_1) - J_{c_\psi}(\tau_0))) = \nabla_\psi J_{c_\psi}(\tau_1) - \frac{\nabla_\psi J_{c_\psi}(\tau_0) \exp(J_{c_\psi}(\tau_0)) + \nabla_\psi J_{c_\psi}(\tau_1) \exp(J_{c_\psi}(\tau_1))}{\exp(J_{c_\psi}(\tau_0)) + \exp(J_{c_\psi}(\tau_1))}$$

$$\nabla_\psi \ln(\sigma(J_{r_\phi}(\tau_0) - J_{r_\phi}(\tau_1))) = 0$$

$$\nabla_\psi \ln(\sigma(J_{r_\phi}(\tau_1) - J_{r_\phi}(\tau_0))) = 0$$

$$\nabla_\psi \ln(\sigma(sJ_{c_\psi}(\tau)))$$

$$= \nabla_\phi \ln(\frac{1}{1 + \exp(sJ_{c_\psi}(\tau))})$$

$$= -\nabla_\phi \ln(1 + \exp(sJ_{c_\psi}(\tau))$$

$$= -\frac{s\nabla_\phi J_{c_\psi}(\tau) \exp(sJ_{c_\psi}(\tau))}{1 + \exp(sJ_{c_\psi}(\tau))}$$

Analogously, we can get

$$\nabla_\phi \ln(\sigma(sJ_{c_\psi}(\tau))) = -\frac{s\nabla_\phi J_{c_\psi}(\tau) \exp(sJ_{c_\psi}(\tau))}{1 + \exp(sJ_{c_\psi}(\tau))} = 0$$

Therefore, we can get

$$
\begin{aligned}
&\nabla_\phi f(\tau_0, \tau_1) \\
&= (1 - y_r)\nabla_\phi \ln(\sigma(J_{r_\phi}(\tau_0) - J_{r_\phi}(\tau_1))) + y_r \nabla_\phi \ln(\sigma(J_{r_\phi}(\tau_1) - J_{r_\phi}(\tau_0))) \\
&+ (1 - y_c)\nabla_\phi \ln(\sigma(J_{c_\psi}(\tau_0) - J_{c_\psi}(\tau_1))) + y_c \nabla_\phi \ln(\sigma(J_{c_\psi}(\tau_1) - J_{c_\psi}(\tau_0))) \\
&+ \nabla_\phi \ln(\sigma(s_0 J_{c_\psi}(\tau_0))) + \nabla_\phi \ln(\sigma(s_1 J_{c_\psi}(\tau_1))) \\
&= (1 - y_r)\nabla_\phi J_{r_\phi}(\tau_0) + y_r \nabla_\phi J_{r_\phi}(\tau_1) - \frac{\nabla_\phi J_{r_\phi}(\tau_0)\exp(J_{r_\phi}(\tau_0)) + \nabla_\phi J_{r_\phi}(\tau_1)\exp(J_{r_\phi}(\tau_1))}{\exp(J_{r_\phi}(\tau_0)) + \exp(J_{r_\phi}(\tau_1))}
\end{aligned}
$$

$$
\begin{aligned}
&\nabla_\psi f(\tau_0, \tau_1) \\
&= (1 - y_r)\nabla_\psi \ln(\sigma(J_{r_\phi}(\tau_0) - J_{r_\phi}(\tau_1))) + y_r \nabla_\psi \ln(\sigma(J_{r_\phi}(\tau_1) - J_{r_\phi}(\tau_0))) \\
&+ (1 - y_c)\nabla_\psi \ln(\sigma(J_{c_\psi}(\tau_0) - J_{c_\psi}(\tau_1))) + y_c \nabla_\psi \ln(\sigma(J_{c_\psi}(\tau_1) - J_{c_\psi}(\tau_0))) \\
&+ \nabla_\psi \ln(\sigma(s_0 J_{c_\psi}(\tau_0))) + \nabla_\psi \ln(\sigma(s_1 J_{c_\psi}(\tau_1))) \\
&= (1 - y_c)\nabla_\phi J_{c_\psi}(\tau_0) + y_c \nabla_\psi J_{c_\psi}(\tau_1) - \frac{\nabla_\psi J_{c_\psi}(\tau_0)\exp(J_{c_\psi}(\tau_0)) + \nabla_\psi J_{c_\psi}(\tau_1)\exp(J_{c_\psi}(\tau_1))}{\exp(J_{c_\psi}(\tau_0)) + \exp(J_{c_\psi}(\tau_1))} \\
&- \frac{s_0 \nabla_\phi J_{c_\psi}(\tau_0)\exp(s_0 J_{c_\psi}(\tau_0))}{1 + \exp(s_0 J_{c_\psi}(\tau_0))} - \frac{s_1 \nabla_\phi J_{c_\psi}(\tau_1)\exp(s_1 J_{c_\psi}(\tau_1))}{1 + \exp(s_1 J_{c_\psi}(\tau_1))}
\end{aligned}
$$

The necessary gradients and elements used for approximating subdifferetial are shown as follows:

$$
\begin{aligned}
&\nabla_\lambda^2 G(\lambda; \phi, \psi) \\
&= \nabla_\lambda J_{c_\psi}(\pi_{\phi,\psi,\lambda}) \\
&= \int_{s \in S} f(\pi_{\phi,\psi,\lambda}, s) \int_{a \in A} \pi_{\phi,\psi,\lambda}(a|s)\nabla_\lambda \ln(\pi_{\phi,\psi,\lambda}(a|s)) J_{c_\psi}(\pi_{\phi,\psi,\lambda}, s, a) da ds
\end{aligned}
$$

$$
\begin{aligned}
&\nabla_{\phi\lambda} G(\lambda; \phi, \psi) \\
&= \nabla_\phi J_{c_\psi}(\pi_{\phi,\psi,\lambda}) \\
&= E^{\pi_{\phi,\psi,\lambda}}[\nabla_\phi \ln(\pi_{\phi,\psi,\lambda}(a|s)) J_{c_\psi}(\pi_{\phi,\psi,\lambda}, s, a)]
\end{aligned}
$$

$$
\begin{aligned}
&\nabla_{\psi\lambda} G(\lambda; \phi, \psi) \\
&= \nabla_\psi J_{c_\psi}(\pi_{\phi,\psi,\lambda}) \\
&= E^{\pi_{\phi,\psi,\lambda}}[\nabla_\psi \ln(\pi_{\phi,\psi,\lambda}(a|s)) J_{c_\psi}(\pi_{\phi,\psi,\lambda}, s, a) + \nabla_\psi c_\psi]
\end{aligned}
$$

Recall that the Lagrangian of the lower-level optimization problem is $L(\phi, \psi, \lambda, \nu) \triangleq G(\lambda; \phi, \psi) - \nu\lambda$. When $\lambda^*(\phi, \psi) = 0$ and $\nu(\phi, \psi) > 0$, the constraint is strictly active, corresponding to the equation (4) in (Xu & Zhu, 2023a), we can get

$$
[\nabla_\phi \lambda^*(\phi, \psi)^T, \nabla_\phi \nu(\phi, \psi)^T]^T = - \begin{bmatrix} \nabla_\lambda^2 L(\phi, \psi, \lambda, \nu) & -\nabla_\lambda \lambda \\ -\nabla_\lambda \lambda & 0 \end{bmatrix}^{-1} [\nabla_{\phi\lambda}^2 L(\phi, \psi, \lambda, \nu)^T, -\nabla_\phi \lambda^T]^T
$$

$$
[\nabla_\phi \lambda^*(\phi, \psi)^T, \nabla_\phi \nu(\phi, \psi)^T]^T = - \begin{bmatrix} \nabla_\lambda^2 G(\lambda; \phi, \psi) & -1 \\ -1 & 0 \end{bmatrix}^{-1} [\nabla_{\phi\lambda}^2 G(\lambda; \phi, \psi)^T, 0]^T,
$$

$$
-[\nabla_{\phi\lambda}^2 G(\lambda; \phi, \psi)^T, 0]^T = \begin{bmatrix} \nabla_\lambda^2 G(\lambda; \phi, \psi) & -1 \\ -1 & 0 \end{bmatrix} [\nabla_\phi \lambda^*(\phi, \psi)^T, \nabla_\phi \nu(\phi, \psi)^T]^T,
$$

from the above equation, we can get $\nabla_\phi \lambda^*(\phi, \psi) = 0$ and $\nabla_\phi \nu(\phi, \psi) = \nabla_{\phi\lambda}^2 G(\lambda; \phi, \psi)$. Similarly, we can get $\nabla_\psi \lambda^*(\phi, \psi) = 0$ and $\nabla_\psi \nu(\phi, \psi) = \nabla_{\psi\lambda}^2 G(\lambda; \phi, \psi)$ through the same process.

When $\lambda^*(\phi, \psi) > 0$ and $\nu(\phi, \psi) = 0$, we can get $\nabla_\phi \nu(\phi, \psi) = 0$, $\nabla_\phi \lambda^*(\phi, \psi) = -\nabla_\lambda^2 G(\lambda; \phi, \psi)^{-1}\nabla_{\phi\lambda}^2 G(\lambda; \phi, \psi)$, $\nabla_\psi \nu(\phi, \psi) = 0$, and $\nabla_\psi \lambda^*(\phi, \psi) = -\nabla_\lambda^2 G(\lambda; \phi, \psi)^{-1}\nabla_{\psi\lambda}^2 G(\lambda; \phi, \psi)$.

## A.5 Proof of Lemma 2

*Proof.* Define $S'_\phi = conv\{H_\phi(\phi'', \psi, \lambda^*(\phi'', \psi)) | \phi'' \in \mathcal{B}(\phi, \epsilon), \lambda^*(\phi'', \psi) \text{ differentiable}\}$. For all $\phi \in \mathbb{R}^{d_\phi}$, consider there exists a small $\epsilon > 0$. Through Taylor's theorem, we can get $|\|H(\phi, \psi, \lambda^*(\phi, \psi))\| - \|H(\phi', \psi, \lambda^*(\phi', \psi))\|| < \mathcal{O}(\epsilon)$ for all $\phi' \in \mathcal{B}(\phi, \epsilon)$. Therefore, we can get $|\|H(\phi, \psi, \lambda^*(\phi, \psi))\| - d(0, S'_\phi)| < \mathcal{O}(\epsilon)$.

From Proposition 4 of (Xu & Zhu, 2023a), we can conclude that $|\|H(\phi', \psi, 0)\| - d(0, S'_{\phi'})| = |\|\nabla_\phi F(\phi', \psi, 0)\| - d(0, S'_\phi)| < \mathcal{O}(\epsilon)$ for all $\phi' \in \mathcal{B}(\phi, \epsilon)$.

For any $g \in S_\phi$, we can conclude $g = \kappa H_\phi(\phi, \psi, \lambda^*(\phi, \psi)) + (1-\kappa)\nabla_\phi F(\phi, \psi, 0)$ with $\kappa \in [0, 1]$. Then we have

$$
\begin{aligned}
|\|g\| - d(0, S'_{\phi'})| &\leq |\kappa\|H_\phi(\phi, \psi, \lambda^*(\phi, \psi))\| + (1-\kappa)\|\nabla_\phi F(\phi, \psi, 0)\| - d(0, S'_\phi)\|| \\
&= |\kappa(\|H_\phi(\phi, \psi, \lambda^*(\phi, \psi))\| - d(0, S'_\phi)) + (1-\kappa)(\|\nabla_\phi F(\phi, \psi, 0)\| - d(0, S'_\phi))\|| \\
&= \kappa\mathcal{O}(\epsilon) + (1-\kappa)\mathcal{O}(\epsilon) \\
&= \mathcal{O}(\epsilon)
\end{aligned}
$$

Finally, we can conclude that $|d(0, S_\phi) - d(0, S'_\phi)| = d(S_\phi, S'_\phi) \leq \mathcal{O}(\epsilon)$.

$\square$

## A.6 Strong Convex

We show that there exists a positive constant $\mu_l$ such that $\nabla_\lambda^2 G(\lambda; \phi, \psi) \geq \mu_l$ for the lower-level problem without constraint. For any lower-level iteration number $k_l$, we can get

$$
\begin{aligned}
&\|\lambda_{k_l+1} - \lambda^*\|^2, \\
&= \|\lambda_{k_l} - \omega_k \hat{\nabla}_\lambda^2 G(\lambda_{k_l}; \phi, \psi) - \lambda^*\|^2, \\
&= \|\lambda_{k_l} - \lambda^*\|^2 + \omega_{k_l}^2\|\hat{\nabla}_\lambda^2 G(\lambda_{k_l}; \phi, \psi)\|^2 - \omega_{k_l}\langle\hat{\nabla}_\lambda^2 G(\lambda_{k_l}; \phi, \psi), \lambda_{k_l} - \lambda^*\rangle, \\
&\leq \|\lambda_{k_l} - \lambda^*\|^2 + \omega_k^2\|Hc_{max}\|^2 + \beta_{k_l}Hc_{max}\|\lambda_0 - \sum_{i=0}^{k_l-1}\omega_i\hat{\nabla}_\lambda^2 G(\lambda_{k_l}; \phi, \psi) - \lambda^*\|, \\
&\leq \|\lambda_{k_l} - \lambda^*\|^2 + \omega_{k_l}^2\|Hc_{max}\|^2 + \omega_{k_l}Hc_{max}(\|\lambda_0 - \lambda^*\| + \|\sum_{i=0}^{k_l-1}\beta_i\hat{\nabla}_\lambda^2 G(\lambda_k; \phi, \psi)\|).
\end{aligned}
\tag{17}
$$

Then we sum both side from $k_l = 0$ to $\bar{k}_l$, we can get

$$
\begin{aligned}
&\sum_{k_l=0}^{\bar{k}_l}\|\lambda_{k_l+1} - \lambda^*\|^2, \\
&\leq \sum_{k_l=0}^{\bar{k}_l}\|\lambda_{k_l} - \lambda^*\|^2 + \beta_{k_l}^2\|Hc_{max}\|^2 + \beta_{k_l}Hc_{max}(\|\lambda_0 - \lambda^*\| + \|\sum_{i=0}^{k_l-1}\beta_i\hat{\nabla}_\lambda^2 G(\lambda_k; \phi, \psi)\|), \\
&\leq \|\lambda_0 - \lambda^*\|^2 + \sum_{i=0}^{k_l-1}\beta_{k_l}^2\|Hc_{max}\|^2 + \beta_{k_l}Hc_{max}(\|\lambda_0 - \lambda^*\| + \|\sum_{i=0}^{k_l-1}\beta_i\hat{\nabla}_\lambda^2 G(\lambda_k; \phi, \psi)\|).
\end{aligned}
\tag{18}
$$

As $\sum_{i=0}^{k_l-1}\omega_{k_l}^2$ and $\sum_{i=0}^{k_l-1}\sum_{i=0}^{k_l-1}\omega_i$ are finite sums, they are bounded. Since the lower-level problem is strictly convex, the value $\lambda^*$ exists, and $\|\lambda_0 - \lambda^*\|$ is bounded. Then, all elements on the right-hand side are bounded. Therefore, $\|\lambda_{\bar{k}_l} - \lambda^*\|^2$ is bounded for any finite $\bar{k}_l \geq 0$. According to the Lemma 9 in (Liu & Zhu, 2022), there exist a positive constant $\mu_l$ such that $\nabla_\lambda^2 G(\lambda; \phi, \psi) \geq \mu_l$ without considering the constraint. As $\lambda \geq 0$ is a convex subset of $\mathbb{R}$, we can conclude that $\nabla_\lambda^2 G(\lambda; \phi, \psi) \geq \mu_l$ when $\lambda$ is differentiable.

## A.7 LIPSCHITZ CONTINUOUS AND BOUNDNESS

Without the constraint, we have the following lemmas.

**Lemma 3.** *Without the constraint, the functions $F(\phi, \psi, \lambda)$ and $G(\lambda; \phi, \psi)$ have the following properties.*
*1. For any $(\phi, \psi, \lambda) \in \mathbb{R}^{d_\phi} \times \mathbb{R}^{d_\psi} \times \mathbb{R}$, the gradients $\nabla_\phi F(\phi, \psi, \lambda)$, $\nabla_\psi F(\phi, \psi, \lambda)$ and $\nabla_\lambda F(\phi, \psi, \lambda)$ are Lipschitz continuous with respect to $\lambda$ with constants $L_{F_\phi} > 0$, $L_{F_\psi} > 0$ and $L_{F_\lambda} > 0$, respectively. The gradient $\nabla_\lambda F(\phi, \psi, \lambda)$ is Lipschitz continuous with respect to $\phi$ and $\psi$ with constants $L'_{F_\lambda} > 0$ and $L''_{F_\lambda} > 0$. The gradients $\nabla_\phi F(\phi, \psi, \lambda)$ is Lipschitz continuous with respect to $\phi$ with constant $L'_{F_\phi}$. The $\nabla_\psi F(\phi, \psi, \lambda)$ is Lipschitz continuous with respect to $\psi$ with constant $L''_{F_\psi}$.*
*2. For any $(\phi, \psi, \lambda) \in \mathbb{R}^{d_\phi} \times \mathbb{R}^{d_\psi} \times \mathbb{R}$, we have $\|\nabla_\lambda F(\phi, \psi, \lambda)\| \le C_{F_\lambda}$ with $C_{F_\lambda} > 0$*
*3. For any $(\phi, \psi, \lambda) \in \mathbb{R}^{d_\phi} \times \mathbb{R}^{d_\psi} \times \mathbb{R}$, the gradients $\nabla_\lambda G(\lambda; \phi, \psi)$, $\nabla_\lambda^2 G(\lambda; \phi, \psi)$, $\nabla_{\phi\lambda}^2 G(\lambda; \phi, \psi)$ and $\nabla_{\psi\lambda}^2 G(\lambda; \phi, \psi)$ are Lipschitz continuous with respect to $\lambda$ with constants $L_{G_\lambda} > 0$, $L_{G_{\lambda\lambda}} > 0$, $L_{G_{\phi\lambda}} > 0$ and $L_{G_{\psi\lambda}}$, respectively. The gradients $\nabla_\lambda^2 G(\lambda; \phi, \psi)$, $\nabla_{\phi\lambda}^2 G(\lambda; \phi, \psi)$ and $\nabla_{\psi\lambda}^2 G(\lambda; \phi, \psi)$ are Lipschitz continuous with respect to $\phi$ with constants $L'_{G_{\lambda\lambda}} > 0$, $L'_{G_{\phi\lambda}} > 0$ and $L'_{G_{\psi\lambda}} > 0$, respectively. The gradients $\nabla_\lambda^2 G(\lambda; \phi, \psi)$, $\nabla_{\phi\lambda}^2 G(\lambda; \phi, \psi)$ and $\nabla_{\psi\lambda}^2 G(\lambda; \phi, \psi)$ are Lipschitz continuous with respect to $\psi$ with constants $L''_{G_{\lambda\lambda}} > 0$, $L''_{G_{\phi\lambda}} > 0$ and $L''_{G_{\psi\lambda}} > 0$, respectively.*
*4. For any $(\phi, \psi, \lambda) \in \mathbb{R}^{d_\phi} \times \mathbb{R}^{d_\psi} \times \mathbb{R}$, we have $\|\nabla_{\phi\lambda}^2 G(\phi, \psi, \lambda)\| \le C_{G_{\phi\lambda}}$ and $\|\nabla_{\psi\lambda}^2 G(\phi, \psi, \lambda)\| \le C_{G_{\psi\lambda}}$ with $C_{G_{\phi\lambda}} > 0$ and $C_{G_{\psi\lambda}} > 0$.*

First, we show the lipschitz continuous of $H_\phi(\phi, \psi, \lambda)$ with respect to $\phi$ and $\lambda$. For any $(\phi, \psi) \in \mathbb{R}^{d_\phi} \times \mathbb{R}^{d_\psi}$, $\lambda_1 \in \mathbb{R}$ and $\lambda_2 \in \mathbb{R}$, we can get

$$
\begin{aligned}
&\|H(\phi, \psi, \lambda_1) - H(\phi, \psi, \lambda_2)\| \\
&= \|\nabla_\phi F(\phi, \psi, \lambda_1) + \nabla_{\phi\lambda}^2 G(\lambda_1; \phi, \psi)[\nabla_\lambda^2 G(\lambda_1; \phi, \psi)]^{-1} \nabla_\lambda F(\phi, \psi, \lambda_1) \\
&\quad - \nabla_\phi F(\phi, \psi, \lambda_2) + \nabla_{\phi\lambda}^2 G(\lambda_2; \phi, \psi)[\nabla_\lambda^2 G(\lambda_2; \phi, \psi)]^{-1} \nabla_\lambda F(\phi, \psi, \lambda_2)\| \\
&\le \|\nabla_\phi F(\phi, \psi, \lambda_1) - \nabla_\phi F(\phi, \psi, \lambda_2)\| \\
&\quad + \|\nabla_{\phi\lambda}^2 G(\lambda_1; \phi, \psi)[\nabla_\lambda^2 G(\lambda_1; \phi, \psi)]^{-1}(\nabla_\lambda F(\phi, \psi, \lambda_1) - \nabla_\lambda F(\phi, \psi, \lambda_2))\| \\
&\quad + \|(\nabla_{\phi\lambda}^2 G(\lambda_1; \phi, \psi) - \nabla_{\phi\lambda}^2 G(\lambda_2; \phi, \psi))[\nabla_\lambda^2 G(\lambda_1; \phi, \psi)]^{-1} \nabla_\lambda F(\phi, \psi, \lambda_2)\| \\
&\quad + \|\nabla_{\phi\lambda}^2 G(\lambda_2; \phi, \psi)([\nabla_\lambda^2 G(\lambda_1; \phi, \psi)]^{-1} - [\nabla_\lambda^2 G(\lambda_2; \phi, \psi)]^{-1})\nabla_\lambda F(\phi, \psi, \lambda_2)\|
\end{aligned}
$$

For any two invertible matrix $X_1$ and $X_2$, we can get $\|X_1^{-1} - X_2^{-1}\| = \|X_1^{-1}(X_1 - X_2)X_2^{-1} \le \|X_1^{-1}\|\|X_1 - X_2\|\|X_2^{-1}\|$. Therefore, keep solving,

$$
\begin{aligned}
&\|H(\phi, \psi, \lambda_1) - H(\phi, \psi, \lambda_2)\| \\
&\le L_{F_\phi}\|\lambda_1 - \lambda_2\| + \frac{L_{F_\lambda}C_{G_{\phi\lambda}}}{\mu_l}\|\lambda_1 - \lambda_2\| + \frac{C_{F_\lambda}L_{G_{\phi\lambda}}}{\mu_l}\|\lambda_1 - \lambda_2\| + \frac{C_{F_\lambda}L_{G_{\lambda\lambda}}C_{G_{\phi\lambda}}}{\mu_l^2}\|\lambda_1 - \lambda_2\| \\
&= (L_{F_\phi} + \frac{L_{F_\lambda}C_{G_{\phi\lambda}}}{\mu_l} + \frac{C_{F_\lambda}L_{G_{\lambda\lambda}}C_{G_{\phi\lambda}}}{\mu_l^2} + \frac{C_{F_\lambda}L_{G_{\lambda\lambda}}C_{G_{\phi\lambda}}}{\mu_l^2})\|\lambda_1 - \lambda_2\|
\end{aligned}
$$

For any $(\psi, \lambda) \in \mathbb{R}^{d_\psi} \times \mathbb{R}$, $\phi_1 \in \mathbb{R}^{d_\phi}$ and $\phi_2 \in \mathbb{R}^{d_\phi}$, we can get

$$\|H(\phi_1, \psi, \lambda_1) - H(\phi_2, \psi, \lambda)\|$$
$$= \|\nabla_\phi F(\phi_1, \psi, \lambda_1) + \nabla^2_{\phi\lambda} G(\lambda; \phi_1, \psi)[\nabla^2_\lambda G(\lambda; \phi_1, \psi)]^{-1} \nabla_\lambda F(\phi_1, \psi, \lambda)$$
$$- \nabla_\phi F(\phi_2, \psi, \lambda_1) + \nabla^2_{\phi\lambda} G(\lambda; \phi_2, \psi)[\nabla^2_\lambda G(\lambda; \phi_2, \psi)]^{-1} \nabla_\lambda F(\phi_2, \psi, \lambda)\|$$
$$\leq \|\nabla_\phi F(\phi_1, \psi, \lambda_1) - \nabla_\phi F(\phi_2, \psi, \lambda_1)\|$$
$$+ \|\nabla^2_{\phi\lambda} G(\lambda; \phi_1, \psi)[\nabla^2_\lambda G(\lambda; \phi_1, \psi)]^{-1} (\nabla_\lambda F(\phi_1, \psi, \lambda) - \nabla_\lambda F(\phi_2, \psi, \lambda))\|$$
$$+ \|(\nabla^2_{\phi\lambda} G(\lambda; \phi_1, \psi) - \nabla^2_{\phi\lambda} G(\lambda; \phi_2, \psi))[\nabla^2_\lambda G(\lambda; \phi_1, \psi)]^{-1} \nabla_\lambda F(\phi_2, \psi, \lambda)\|$$
$$+ \|\nabla^2_{\phi\lambda} G(\lambda; \phi_2, \psi)([\nabla^2_\lambda G(\lambda; \phi_1, \psi)]^{-1} - [\nabla^2_\lambda G(\lambda; \phi_2, \psi)]^{-1}) \nabla_\lambda F(\phi_2, \psi, \lambda)\|$$
$$\leq L'_{F_\phi} \|\phi_1 - \phi_2\| + \frac{L'_{F_\lambda} C_{G_{\phi\lambda}}}{\mu_l} \|\phi_1 - \phi_2\| + \frac{C_{F_\lambda} L'_{G_{\phi\lambda}}}{\mu_l} \|\phi_1 - \phi_2\| + \frac{C_{F_\lambda} L'_{G_{\lambda\lambda}} C_{G_{\phi\lambda}}}{\mu_l^2} \|\phi_1 - \phi_2\|$$
$$= (L'_{F_\phi} + \frac{L'_{F_\lambda} C_{G_{\phi\lambda}}}{\mu_l} + \frac{C_{F_\lambda} L'_{G_{\phi\lambda}}}{\mu_l} + \frac{C_{F_\lambda} L'_{G_{\lambda\lambda}} C_{G_{\phi\lambda}}}{\mu_l^2}) \|\phi_1 - \phi_2\|$$

Let $L_\lambda = L_{F_\phi} + \frac{L_{F_\lambda} C_{G_{\phi\lambda}}}{\mu_l} + \frac{C_{F_\lambda} L_{G_{\lambda\lambda}} C_{G_{\phi\lambda}}}{\mu_l^2} + \frac{C_{F_\lambda} L_{G_{\lambda\lambda}} C_{G_{\phi\lambda}}}{\mu_l^2}$ and $L_\phi = L'_{F_\phi} + \frac{L'_{F_\lambda} C_{G_{\phi\lambda}}}{\mu_l} + \frac{C_{F_\lambda} L'_{G_{\phi\lambda}}}{\mu_l} + \frac{C_{F_\lambda} L'_{G_{\lambda\lambda}} C_{G_{\phi\lambda}}}{\mu_l^2}$, we have $\|H(\phi, \psi, \lambda_1) - H(\phi, \psi, \lambda_2)\| \leq L_\lambda \|\lambda_1 - \lambda_2\|$ and $\|H(\phi_1, \psi, \lambda_1) - H(\phi_2, \psi, \lambda)\| \leq L_\phi \|\phi_1 - \phi_2\|$. Next, we have

$$\|H(\phi_1, \psi, \lambda^*(\phi_1, \psi))) - H(\phi_2, \psi, \lambda^*(\phi_2, \psi))\|$$
$$\leq \|H(\phi_1, \psi, \lambda^*(\phi_1, \psi))) - H(\phi_1, \psi, \lambda^*(\phi_2, \psi))\| + \|H(\phi_1, \psi, \lambda^*(\phi_2, \psi))) - H(\phi_2, \psi, \lambda^*(\phi_2, \psi))\|$$
$$\leq L_\lambda \|\lambda^*(\phi_1, \psi) - \lambda^*(\phi_2, \psi)\| + L_\phi \|\phi_1 - \phi_2\|$$
$$\leq (\frac{L_\lambda C_{G_{\phi\lambda}}}{\mu_g} + L_\phi) \|\phi_1 - \phi_2\|$$

Let $L = \frac{L_\lambda C_{G_{\phi\lambda}}}{\mu_g} + L_\phi$, we can conclude that $\|H(\phi_1, \psi, \lambda^*(\phi_1, \psi))) - H(\phi_2, \psi, \lambda^*(\phi_2, \psi))\| \leq L \|\phi_1 - \phi_2\|$.

## A.8 PROOF OF THEOREM 1

First, we need to quantify all approximation errors. The overall approximation error comes from two aspects, one from the approximation of the subgradient and another one is the difference between the calculated $\lambda$ and $\lambda^*$ from the lower-level. The approximation of the subgradient is $\mathcal{O}(\epsilon)$ as shown in Lemma 2. Following the idea of gradient descent, we can show that $\|\lambda_{k_l} - \lambda^*\| \leq (1 - \frac{L_{G_\lambda}}{\mu_g})^{k_l} \|\lambda_0 - \lambda^*\|$. Therefore, we conclude the approximation error at iteration $k$ as $E_k = C_s \epsilon + (1 - \frac{L_{G_\lambda}}{\mu_g})^{k_l} \|\lambda_0 - \lambda^*\|$, where $C_s$ is coefficient of the approximation error of the subgradient. Next, we can show

$$F(\phi_{k+1}, \psi, \lambda^*(\phi_{k+1}, \psi))$$
$$\leq F(\phi_k, \psi, \lambda^*(\phi_k, \psi)) + \langle H'(\phi_k, \psi, \lambda^*(\phi_1, \psi)), \phi_{k+1} - \phi_k \rangle + \frac{L}{2} \|\phi_{k+1} - \phi_k\|^2$$
$$= F(\phi_k, \psi, \lambda^*(\phi_k, \psi)) - \alpha_k \langle H'(\phi_k, \psi, \lambda^*(\phi_1, \psi)), H(\phi_k, \psi, \lambda^*(\phi_1, \psi)) \rangle + \frac{L\alpha_k^2}{2} \|H(\phi_k, \psi, \lambda^*(\phi_1, \psi))\|^2$$

Define $H(\phi_k, \psi, \lambda^*(\phi_1, \psi)) = H'(\phi_k, \psi, \lambda^*(\phi_1, \psi)) + E_k$. Then the expectation of $F(\phi_{k+1}, \psi, \lambda^*(\phi_{k+1}, \psi))$ becomes:

$$E[F(\phi_{k+1}, \psi, \lambda^*(\phi_{k+1}, \psi))]$$
$$\leq F(\phi_k, \psi, \lambda^*(\phi_k, \psi)) - \alpha_k \langle H'(\phi_k, \psi, \lambda^*(\phi_k, \psi)), H'(\phi_k, \psi, \lambda^*(\phi_k, \psi)) + E_k \rangle$$
$$+ \frac{L\alpha_k^2}{2}\|H'(\phi_k, \psi, \lambda^*(\phi_k, \psi)) + H(\phi_k, \psi, \lambda^*(\phi_k, \psi) - H'(\phi_k, \psi, \lambda^*(\phi_k, \psi)\|^2$$
$$\leq F(\phi_k, \psi, \lambda^*(\phi_k, \psi)) - \alpha_k \langle H'(\phi_k, \psi, \lambda^*(\phi_1, \psi)), H'(\phi_k, \psi, \lambda^*(\phi_k, \psi)) + E_k \rangle$$
$$+ \frac{L\alpha_k^2}{2}Var(H(\phi_k, \psi, \lambda^*(\phi_k, \psi))) + \frac{L\alpha_k^2}{2}\|H'(\phi_k, \psi, \lambda^*(\phi_k, \psi))\|^2 + L\alpha_k^2\langle H'(\phi_k, \psi, \lambda^*(\phi_1, \psi)), E_k \rangle$$
$$= F(\phi_k, \psi, \lambda^*(\phi_k, \psi)) - (\alpha_k - \frac{L\alpha_k^2}{2})\|H'(\phi_k, \psi, \lambda^*(\phi_1, \psi))\|^2$$
$$- (\alpha_k - L\alpha_k^2)\langle H'(\phi_k, \psi, \lambda^*(\phi_k, \psi)), E_k \rangle + \frac{L\alpha_k^2}{2}Var(H(\phi_k, \psi, \lambda^*(\phi_k, \psi))) + \frac{L\alpha_k^2}{2}\|E_k\|$$

By using the fact that $2\langle H'(\phi_k, \psi, \lambda^*(\phi_K, \psi)), E_k \rangle \leq \|H'(\phi_k, \psi, \lambda^*(\phi_1, \psi))\|^2 + \|E_k\|^2$ and choosing $\alpha_k < \frac{1}{L}$, we can get

$$E[F(\phi_{k+1}, \psi, \lambda^*(\phi_{k+1}, \psi))]$$
$$\leq F(\phi_k, \psi, \lambda^*(\phi_k, \psi)) - \frac{\alpha_k}{2}(\|H'(\phi_k, \psi, \lambda^*(\phi_k, \psi))\|^2 - \|E_k\|^2) + \frac{L\alpha_k^2}{2}Var(H(\phi_k, \psi, \lambda^*(\phi_k, \psi)))$$

Then consider from $k = 0$, we can get

$$\sum_{k=0}^{K-1} \frac{\alpha_k}{2} E[\|H'(\phi_k, \psi, \lambda^*(\phi_k, \psi))\|^2]$$
$$\leq F(\phi_0, \psi, \lambda^*(\phi_0, \psi)) - F^* + \sum_{k=0}^{K-1} \frac{\alpha_k}{2}\|E_k\|^2 + \frac{L\alpha_k^2}{2}Var(H(\phi_k, \psi, \lambda^*(\phi_k, \psi)))$$

Here, we quantify the value of $Var(H(\phi_k, \psi, \lambda^*(\phi_k, \psi)))$.

$$Var(H(\phi_k, \psi, \lambda^*(\phi_k, \psi)))$$
$$\leq (\|H(\phi_k, \psi, \lambda^*(\phi_k, \psi)))\|^2$$
$$= \|H'(\phi_k, \psi, \lambda^*(\phi_k, \psi))) + C_s\epsilon\|^2$$
$$\leq (C_{F_\phi} + \frac{C_{G_{\phi\lambda}}C_F}{\mu_g} + C_s\epsilon)^2$$

Then we can conclude that

$$\frac{1}{K}\sum_{k=0}^{K-1} E[\|H'(\phi_k, \psi, \lambda^*(\phi_k, \psi))\|^2]$$
$$\leq \frac{2}{K\alpha_k}(F(\phi_0, \psi, \lambda^*(\phi_0, \psi)) - F^*) + \frac{1}{K}\sum_{k=0}^{K-1}\|E_k\|^2 + L\alpha_k Var(H(\phi_k, \psi, \lambda^*(\phi_k, \psi)))$$
$$\leq \frac{2}{K\alpha_k}(F(\phi_0, \psi, \lambda^*(\phi_0, \psi)) - F^*) + \frac{1}{K}\sum_{k=0}^{K-1}C_s^2\epsilon^2 + 2C_s\epsilon(1 - \frac{L_{G_\lambda}}{\mu_g})^{k_l}\|\lambda_0 - \lambda^*\| + ((1 - \frac{L_{G_\lambda}}{\mu_g})^{k_l}\|\lambda_0 - \lambda^*\|)^2$$
$$+ L\alpha_k((C_{F_\phi} + \frac{C_{G_{\phi\lambda}}C_F}{\mu_g})^2 + 2C_s\epsilon(C_{F_\phi} + \frac{C_{G_{\phi\lambda}}C_F}{\mu_g}) + C_s^2\epsilon^2)$$

By choosing $\alpha_k = \frac{1}{L\sqrt{K}}$, $k_l = k$ and $\epsilon = \frac{1}{k^2}$, we can get

$$\frac{1}{K}\sum_{k=0}^{K-1} E[\|H'(\phi_k, \psi, \lambda^*(\phi_k, \psi))\|^2]$$

$$\leq \frac{2L}{\sqrt{K}}(F(\phi_0, \psi, \lambda^*(\phi_0, \psi)) - F^*) + \frac{C_s^2\pi^4}{90K} + \frac{2C_s(\mu_g - L_{G_\lambda})\|\lambda_0 - \lambda^*\|\pi^2}{6LK} + \frac{(\mu_g - L_{G_\lambda})\|\lambda_0 - \lambda^*\|^2}{LK}$$

$$+ \frac{(C_{F_\phi} + \frac{C_{G_{\phi\lambda}}C_F}{\mu_g})^2}{\sqrt{K}} + \frac{2C_s\pi^2(C_{F_\phi} + \frac{C_{G_{\phi\lambda}}C_F}{\mu_g})}{6\sqrt{K}} + \frac{c_s^2\pi^4}{90\sqrt{K}}$$

$$= \frac{C_1}{\sqrt{K}} + \frac{C_2}{K}$$

where $C_1 = 2L(F(\phi_0, \psi, \lambda^*(\phi_0, \psi)) - F^*) + (C_{F_\phi} + \frac{C_{G_{\phi\lambda}}C_F}{\mu_g})^2 + \frac{2C_s\pi^2(C_{F_\phi} + \frac{C_{G_{\phi\lambda}}C_F}{\mu_g})}{6} + \frac{c_s^2\pi^4}{90}$

and $C_2 = \frac{C_s^2\pi^4}{90} + \frac{2C_s(\mu_g - L_{G_\lambda})\|\lambda_0 - \lambda^*\|\pi^2}{6L} + \frac{(\mu_g - L_{G_\lambda})\|\lambda_0 - \lambda^*\|^2}{L}$. Then we can conclude that

$$\min\{\|g_\phi\|^2 | g_\phi \in S_\phi\} \leq \frac{1}{K}\sum_{k=0}^{K-1} E[\|H'(\phi_k, \psi, \lambda^*(\phi_k, \psi))\|^2] \leq \mathbb{O}(\frac{1}{\sqrt{K}}) \tag{19}$$

### A.9 PROOF OF THEOREM 2

**Assumption 3.** *The human preference models $h_{r_h}(\cdot|\tau_0, \tau_1) \propto \exp J_{r_h}(\tau_p)$ and $h_{c_h}(\cdot|\tau_0, \tau_1) \propto \exp J_{c_h}(\tau_p)$ where $\tau_p$ is the preferred trajectory between $\tau_0$ and $\tau_1$.*

*Proof.* According to the assumption 3, for each trajectory pair $\Gamma = \{\tau_0, \tau_1\}$, the human preference is generated according to the human preference model $h_{r_h}(\tau|\Gamma)$:

$$h_{r_h}(\tau|\Gamma) = \frac{\exp(J_{r_h}(\tau))}{\sum_{\tau' \in \Gamma} \exp(J_{r_h}(\tau'))}$$

where $r_h$ is the reward function of the human.

Define a preference model class $\mathcal{H}(\Gamma)$ defined on $\Gamma$, which is parametrized by the function $r_\phi$:

$$\mathcal{H}(\Gamma) = \{h_{r_\phi}(\tau|\Gamma) = \frac{\exp(J_{r_\phi}(\tau))}{\sum_{\tau' \in \Gamma} \exp(J_{r_\phi}(\tau'))} \quad \text{for some} \quad r_\phi \in \mathcal{F}\}.$$

Define $N_{[]}(\mathcal{H}(\Gamma), \|\cdot\|_\infty, \frac{1}{n})$ as the bracketing number of $\Pi(\Gamma)$.

From the upper-level objective function, we can get

$$\frac{1}{n}\sum_{i=1}^n \ln(\frac{\exp(J_{r_\phi}(\tau))}{\sum_{\tau' \in \Gamma} \exp(J_{r_\phi}(\tau'))}) \geq \frac{1}{n}\sum_{i=1}^n \ln(\frac{\exp(J_{r_h}(\tau))}{\sum_{\tau' \in \Gamma} \exp(J_{r_h}(\tau'))})$$

$$\frac{1}{n}\sum_{i=1}^n \ln(h_{r_\phi}(\tau|\Gamma)) \geq \frac{1}{n}\sum_{i=1}^n \ln(h_{r_h}(\tau|\Gamma))$$

$$\frac{1}{n}\sum_{i=1}^n \ln(h_{r_\phi}(\tau|\Gamma)) - \frac{1}{n}\sum_{i=1}^n \ln(h_{r_h}(\tau|\Gamma)) \geq 0$$

$$\frac{1}{n}\sum_{i=1}^n \ln(\frac{h_{r_\phi}(\tau|\Gamma)}{h_{r_h}(\tau|\Gamma)}) \geq 0.$$

For any $\hat{h}$ belongs to the same bracket with $h_{r_h}$, according to Markov's inequality, we can get

$$P(\exp(\sum_{i=1}^n \frac{1}{2}\ln(\frac{\hat{h}}{h_{r_h}})) \geq \exp(\epsilon)) \leq \frac{E_{\pi, h_{r_h}}[\exp(\sum_{i=1}^n \frac{1}{2}\ln(\frac{\hat{h}}{h_{r_h}}))]}{\exp(\epsilon)}.$$

Through Boole's inequality, we can get

$$P(\forall \hat{h} \in \Pi, \exp(\sum_{i=1}^{n} \frac{1}{2}\ln(\frac{\hat{h}}{h_{r_h}})) \geq \exp(\epsilon)) \leq \frac{E_{\pi,h_{r_h}}[\exp(\sum_{i=1}^{n} \frac{1}{2}\ln(\frac{\hat{h}}{h_{r_h}}))] \cdot N_{[]}(\mathcal{H}(\Gamma), \|\cdot\|_\infty, \frac{1}{n})}{\exp(\epsilon)}.$$

Therefore, with probability at least $\delta$, we can get

$$\delta = \frac{E_{\pi,h_{r_h}}[\exp(\sum_{i=1}^{n} \frac{1}{2}\ln(\frac{\hat{h}}{h_{r_h}}))] \cdot N_{[]}(\mathcal{H}(\Gamma), \|\cdot\|_\infty, \frac{1}{n})}{\exp(\epsilon)},$$

$$\epsilon = \ln(\frac{E_{\pi,h_{r_h}}[\exp(\sum_{i=1}^{n} \frac{1}{2}\ln(\frac{\hat{h}}{h_{r_h}}))] \cdot N_{[]}(\mathcal{H}(\Gamma), \|\cdot\|_\infty, \frac{1}{n})}{\delta}$$

$$\epsilon = n\ln(E_{\pi,h_{r_h}}[\exp(\frac{1}{2}\ln(\frac{\hat{h}}{h_{r_h}}))]) + \ln(\frac{N_{[]}(\mathcal{H}(\Gamma), \|\cdot\|_\infty, \frac{1}{n})}{\delta}).$$

With probability at least $1 - \delta$ and any policy $\pi$ we can get

$$\sum_{i=1}^{n} \frac{1}{2}\ln(\frac{\hat{h}}{h_{r_h}}) \leq n\ln(E_{\pi,h_{r_h}}[\exp(\frac{1}{2}\ln(\frac{\hat{h}}{h_{r_h}}))]) + \ln(\frac{N_{[]}(\mathcal{H}(\Gamma), \|\cdot\|_\infty, \frac{1}{n})}{\delta})$$

$$0 \leq n\ln(E_{\pi,h_{r_h}}[\exp(\frac{1}{2}\ln(\frac{\hat{h}}{h_{r_h}}))]) + \ln(\frac{N_{[]}(\mathcal{H}(\Gamma), \|\cdot\|_\infty, \frac{1}{n})}{\delta})$$

$$\leq n\ln(E_{\pi,h_{r_h}}[\sqrt{\frac{\hat{h}}{h_{r_h}}})]) + \ln(\frac{N_{[]}(\mathcal{H}(\Gamma), \|\cdot\|_\infty, \frac{1}{n})}{\delta})$$

$$= n\ln(E_\pi[\sqrt{\hat{h}h_{r_h}}]) + \ln(\frac{N_{[]}(\mathcal{H}(\Gamma), \|\cdot\|_\infty, \frac{1}{n})}{\delta}).$$

Through $\ln(x) \leq x - 1$, we can get

$$1 - E_\pi[\sqrt{\hat{h}h_{r_h}}] \leq \frac{1}{n}\ln(\frac{N_{[]}(\mathcal{H}(\Gamma), \|\cdot\|_\infty, \frac{1}{n})}{\delta}).$$

Next, we can bound the difference between $\hat{h}$ and $h_{r_h}$.

$$E_\pi[\|\hat{h} - h_{r_h}\|_1^2] = E_\pi[\|(\hat{h}^{\frac{1}{2}} - h_{r_h}^{\frac{1}{2}})(\hat{h}^{\frac{1}{2}} + h_{r_h}^{\frac{1}{2}})\|_1^2]$$

$$\leq E_\pi[\|\hat{h}^{\frac{1}{2}} - h_{r_h}^{\frac{1}{2}}\|_1^2] E_\pi[\|\hat{h}^{\frac{1}{2}} + h_{r_h}^{\frac{1}{2}}\|_1^2]$$

$$\leq (2(1 - E_\pi[\sqrt{\hat{h}h_{r_h}}]) + \frac{1}{n})(2E_\pi[\hat{h} + h_{r_h}])$$

$$\leq (\frac{2}{n}\ln(\frac{N_{[]}(\mathcal{H}(\Gamma), \|\cdot\|_\infty, \frac{1}{n})}{\delta}) + \frac{1}{n})(\frac{2}{n} + 4)$$

$$\leq \mathcal{O}(\frac{1}{n}\ln(\frac{N_{[]}(\mathcal{H}(\Gamma), \|\cdot\|_\infty, \frac{1}{n})}{\delta}))$$

Then we can bound the difference between $h_{r_\phi}$ and $h_{r_h}$.

$$E_\pi[\|h_{r_\phi} - h_{r_h}\|_1^2] = E_\pi[\|h_{r_\phi} - \hat{h} + \hat{h} - h_{r_h}\|_1^2]$$

$$\leq E_\pi[\|h_{r_\phi} - \hat{h}\|_1^2] + E_\pi[\|\hat{h} - h_{r_h}\|_1^2]$$

$$\leq \mathcal{O}(\frac{1}{n}\ln(\frac{N_{[]}(\mathcal{H}(\Gamma), \|\cdot\|_\infty, \frac{1}{n})}{\delta}))$$

$$\|\hat{h} - h_{r_h}\|_1^2 - \|h_{r_\phi} - h_{r_h}\|_1^2 \leq (|\hat{h} - h_{r_h}| + |h_{r_\phi} - h_{r_h}|)(|\hat{h} - h_{r_h}| - |h_{r_\phi} - h_{r_h}|)$$

$$\leq (4 + \frac{1}{n})\frac{1}{n}.$$

We can conclude that

$$E_\pi[\|h_{r_\phi} - h_{r_h}\|_1^2] \leq \mathcal{O}(\frac{1}{n}\ln(\frac{N_{[]}(\mathcal{H}(\Gamma), \|\cdot\|_\infty, \frac{1}{n})}{\delta}))$$

Since the softmax function is continuous, there exists a constant $L_h \geq 0$ such that

$$|h_{r_\phi} - h_{r_h}| \leq L_h \|r_\phi - r_h\|_\infty.$$

Then, according to Lemma 2.14 in (Sen, 2018), we can get $N_{[]}(\mathcal{H}(\Gamma), \|\cdot\|_\infty, \frac{1}{n}) \leq N(\mathcal{F}, \|\cdot\|_\infty, \frac{1}{4n})$ for all $K$ iterations. Overall, we can get

$$E_\pi[\|h_{r_\phi} - h_{r_h}\|_1^2] \leq \mathcal{O}(\frac{1}{n} \ln(\frac{N(\mathcal{F}, \|\cdot\|_\infty, \frac{1}{n})}{\delta})).$$

Meanwhile, define $c_{h_{r_h}} \triangleq \frac{\exp(-Hf_b) + \exp(Hf_b)}{\exp(-Hf_b)}$ is the upper bound of the $\frac{1}{h_{r_h}}$ we have

$$E_\pi[\|J_{r_\phi}(\tau) - J_{r_h}(\tau)\|_1^2] = E_\pi[\|\ln(\frac{h_{r_\phi}}{h_{r_h}})\|_1^2]$$

$$\leq E_\pi[\|c_{h_{r_h}}(h_{r_\phi} - h_{r_h})\|_1^2]$$

$$\leq \mathcal{O}(\frac{1}{n} \ln(\frac{N(\mathcal{F}, \|\cdot\|_\infty, \frac{1}{n})}{\delta})).$$

From above, we can get $\|J_{r_\phi}(\pi) - J_{r_h}(\pi)\|_1^2 \leq \mathcal{O}(\frac{1}{n} \ln(\frac{N(\mathcal{F}, \|\cdot\|_\infty, \frac{1}{n})}{\delta}))$ for all $\pi$. Let $\pi^*$ be the policy of the human, we can conclude that

$$SubOptR(\pi_{\phi^*, \psi^*, \lambda^*(\phi^*, \psi^*)}) = J_{r_h}(\pi^*) - J_{r_h}(\pi_{\phi^*, \psi^*, \lambda^*(\phi^*, \psi^*)})$$

$$= J_{r_h}(\pi^*) - J_{r_{\phi^*}}(\pi^*) + J_{r_{\phi^*}}(\pi^*) - J_{r_{\phi^*}}(\pi_{\phi^*, \psi^*, \lambda^*(\phi^*, \psi^*)})$$

$$+ J_{r_{\phi^*}}(\pi_{\phi^*, \psi^*, \lambda^*(\phi^*, \psi^*)}) - J_{r_h}(\pi_{\phi^*, \psi^*, \lambda^*(\phi^*, \psi^*)})$$

Since $\pi_{\phi^*, \psi^*, \lambda^*(\phi^*, \psi^*)}$ is the optimal policy corresponding to $r_{\phi^*}$, we can get $J_{r_{\phi^*}}(\pi^*) - J_{r_{\phi^*}}(\pi_{\phi^*, \psi^*}) \leq 0$. Meanwhile, $\|J_{r_h}(\pi^*) - J_{r_{\phi^*}}(\pi^*)\|_1^2 \leq \mathcal{O}(\frac{1}{n} \ln(\frac{N(\mathcal{F}, \|\cdot\|_\infty, \frac{1}{n})}{\delta}))$ and $\|J_{r_{\phi^*}}(\pi_{\phi^*, \psi^*}) - J_{r_h}(\pi_{\phi^*, \psi^*})\|_1^2 \leq \mathcal{O}(\frac{1}{n} \ln(\frac{N(\mathcal{F}, \|\cdot\|_\infty, \frac{1}{n})}{\delta}))$. Therefore, we can conclude that

$$SubOptR(\pi_{\phi^*, \psi^*, \lambda^*(\phi^*, \psi^*)}) \leq J_{r_h}(\pi^*) - J_{r_{\phi^*}}(\pi^*) + J_{r_{\phi^*}}(\pi_{\phi^*, \psi^*, \lambda^*(\phi^*, \psi^*)}) - J_{r_h}(\pi_{\phi^*, \psi^*, \lambda^*(\phi^*, \psi^*)})$$

$$\leq \mathcal{O}(\sqrt{\frac{1}{n} \ln(\frac{N(\mathcal{F}, \|\cdot\|_\infty, \frac{1}{n})}{\delta})})$$

The proof process for $c_\psi$ is the same as that of $r_\phi$. $\square$

### A.10 EXPERIMENT DETAILS

All Python3 codes are run on a Windows 10 desktop with 13th Gen Intel(R) Core(TM) i7-13700KF CPU and 32 GB of RAM. In CB-RLHF, the reward function updates every 10 environment set with three sets of feedback. We use SAC to calculate the policy, and the hyperparameters of SAC are shown below.

| Hyperparameter | Value |
|---|---|
| Initial temperature | 0.2 |
| Learning rate | 0.001 |
| Critic target update freq | 10 |
| $(\beta_1, \beta_2)$ | (.9, .999) |
| Hidden units per each layer | 256 |
| Batch Size | 100 |
| Optimizer | Adam |
| Critic EMA $\tau$ | 0.005 |
| Discount $\gamma$ | .99 |

### A.10.1 WLAKER2D

The walker is a two-dimensional bipedal robot consisting of a torso and two legs, each with a thigh, shin, and foot joint. Its movement is controlled by applying torques to these joints. The continuous

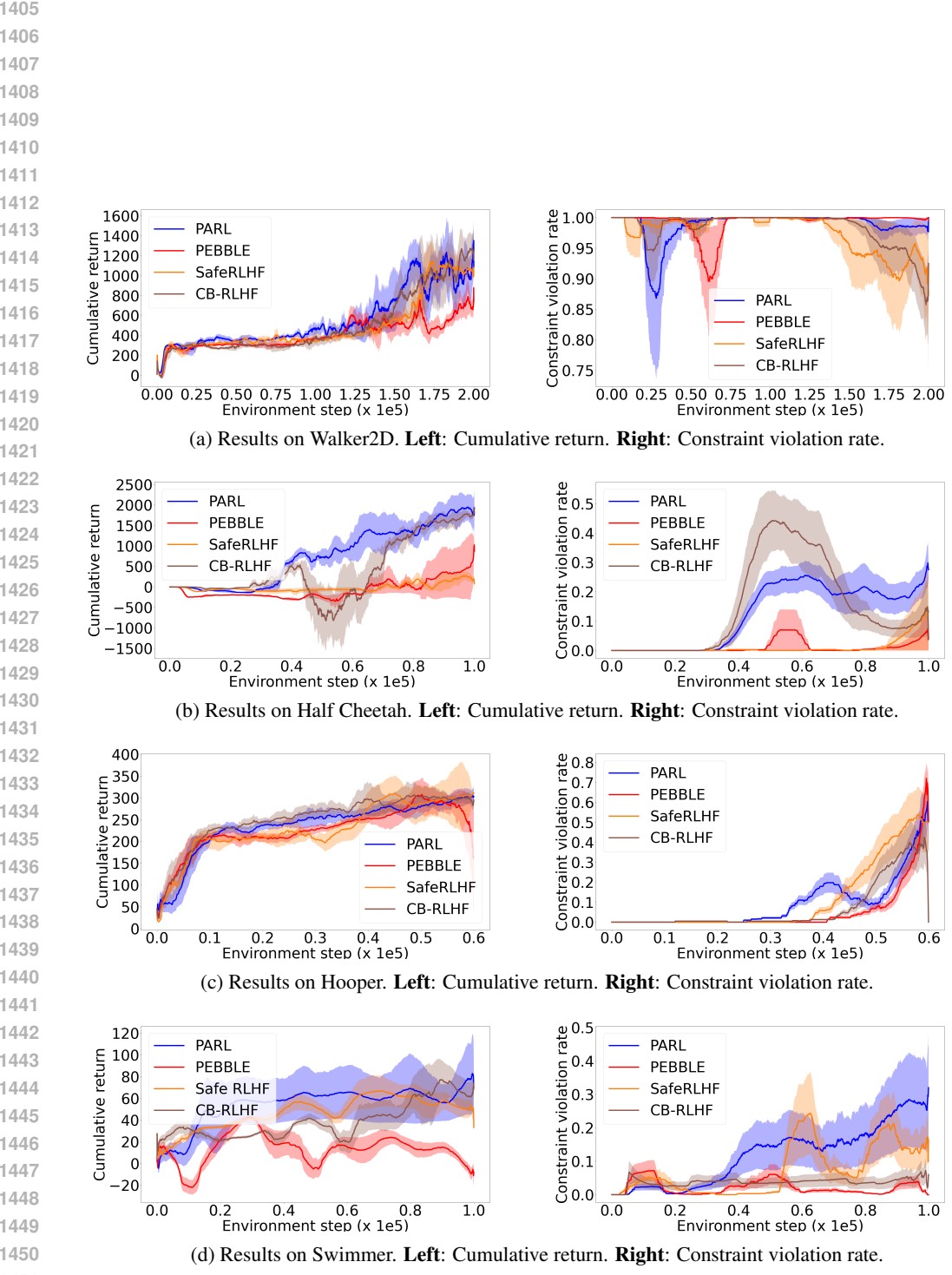

(a) Results on Walker2D. **Left**: Cumulative return. **Right**: Constraint violation rate.

(b) Results on Half Cheetah. **Left**: Cumulative return. **Right**: Constraint violation rate.

(c) Results on Hooper. **Left**: Cumulative return. **Right**: Constraint violation rate.

(d) Results on Swimmer. **Left**: Cumulative return. **Right**: Constraint violation rate.

observation and action spaces have dimensions of 17 and 6, respectively. The walker's objective is to move forward as quickly as possible while maintaining stability. The default reward function is the sum of a healthy reward, a forward reward, and a control cost. The healthy reward is a fixed value of 1 per timestep if the walker remains upright. The forward reward is proportional to the forward displacement, given by $125(x - x')$, where $x$ and $x'$ denote the walker's positions after and before the action, respectively. The control cost is $0.001\|a_w\|_2^2$, where $a_w$ is the action. In addition, we impose a constraint on the angles of the torso and six joints, applying penalties when their absolute values exceed 0.75 radians. The cost function is $\sum_{i=1}^{7} \max(0, |ag_i| - 0.75)$, where $ag_i$ denotes the angle of the torso or a joint.

### A.10.2 HALF CHEETAH

The Half Cheetah is a 2-dimensional robot with 8 joints. Its movement is controlled by applying torques to these joints. The continuous observation and action spaces have dimensions of 17 and 6, respectively. The robot's objective is to move forward as quickly as possible. The default reward function is the sum of a forward reward, and a control cost. The forward reward is proportional to the forward displacement, given by $20(x - x')$, where $x$ and $x'$ denote the walker's positions after and before the action, respectively. The control cost is $0.001\|a_w\|_2^2$, where $a_w$ is the action. In addition, we impose a constraint on the angular velocity of 7 hinges, applying penalties when their absolute values exceed 12 radians per second. The cost function is $\sum_{i=1}^{7} \max(0, |av_i| - 12)$, where $av_i$ denotes the angular velocity of a hinge.

### A.10.3 HOOPER

The hopper is a two-dimensional one-legged figure 4 body parts: torso, thigh, leg, and foot. Its movement is controlled by applying torques to three hinges. The continuous observation and action spaces have dimensions of 11 and 3, respectively. The robot's objective is to move forward as quickly as possible. The default reward function is the same as that of the Walker2D. In addition, we impose a constraint on the torques of hinges, applying penalties when their absolute values exceed 0.75 Nm. The cost function is $\sum_{i=1}^{3} \max(0, |t_i| - 0.75)$, where $t_i$ denotes the torque of a hinge.

### A.10.4 SWIMMER

The swimmer is a planar 2D agent with a torso and two linked segments. Its movement is controlled by applying torques to the rotors and using fluid friction. The continuous observation and action spaces have dimensions of 8 and 2, respectively. The robot's objective is to move as fast as possible towards the right. The default reward function is the sum of a forward reward, and a control cost. The forward reward is proportional to the forward displacement, given by $25(x - x')$, where $x$ and $x'$ denote the walker's positions after and before the action, respectively. The control cost is $0.0001\|a_w\|_2^2$, where $a_w$ is the action. In addition, we impose a constraint on the angular velocity of 2 hinges, applying penalties when their absolute values exceed 0.75 radians per second. The cost function is $\sum_{i=1}^{2} \max(0, |t_i| - 0.75)$, where $t_i$ denotes the the torque of a hinge.

### A.11 THE USE OF LARGE LANGUAGE MODELS (LLMS)

We confirm that LLM (ChatGPT 5) assistance was limited to improving grammar and readability.

