# OpenReview forum: "A Constrained Bi-level Optimization Framework for Constrained Reinforcement Learning from Human Feedback"
_ICLR.cc/2026/Conference — Submitted to ICLR 2026_

### Official Review · Reviewer_mhU2 · 2025-10-21

**Soundness:** 2
**Presentation:** 2
**Contribution:** 2
**Rating:** 4
**Confidence:** 3

**Summary:**

This paper addresses the problem of learning from human feedback in reinforcement learning (RL) scenarios that involve constraints. The authors formulate this problem as a constrained bi-level optimization task. In this formulation, the upper-level problem learns reward and cost functions from human feedback, while the lower-level problem solves for the optimal policy under those given functions.
To solve this, the paper introduces an algorithm called CB-RLHF. The algorithm handles the non-convexity of the lower-level problem by using its dual formulation and addresses potential non-differentiability using a Clarke subdifferential approximation. The authors provide a theoretical convergence guarantee and present pseudo experiments on MuJoCo environments to show that the method can address both misalignment and constraint inference limitations.

**Strengths:**

- Important Problem Formulation: The paper tackles the important and practical problem of learning from human feedback while simultaneously respecting safety or ethical constraints. The formulation of constrained bi-level optimization problem might be a notable contribution.
- Theoretical Grounding: The authors make a serious attempt to provide a theoretical foundation for their algorithm. They provide proofs for the convergence of their method, which adds rigor to their claims.

**Weaknesses:**

Insufficient Experimental Validation: The experiments are not sufficient to support the paper's claims about RLHF.
- Unrealistic Oracle: The experiments use a synthetic oracle based on ground-truth functions, which bypasses the core challenges of noisy, ambiguous, and costly feedback from real humans.
- Missing Key Baseline: The paper fails to compare against the most obvious baseline: a standard, iterative RLHF approach where the update cycle is simply run much more frequently. It is unclear if the proposed complex optimization is better than just iterating faster.
- Lack of Ablation Study: The method introduces two key components (a cost function and a bi-level framework) but provides no ablation study to disentangle their effects. We cannot know what is truly responsible for the performance changes.
- Lack of Analysis: The paper presents results but offers almost no analysis. It claims success even when the method underperforms baselines on some metrics (e.g., Hopper return), with no discussion as to why.

Lack of Clarity in Theoretical Justification: The lack of clarity makes the core justification for the method confusing.

**Questions:**

Could the authors clarify how strong duality (Lemma 1) is guaranteed to hold for the problems?

---

> ### Author Response · Authors · 2025-11-28
>
> > Unrealistic Oracle: The experiments use a synthetic oracle based on ground-truth functions, which bypasses the core challenges of noisy, ambiguous, and costly feedback from real humans.
>
> **Answer:** We thank the reviewer for raising this important point. The purpose of not including noise in the labels is to isolate and directly evaluate the optimization behavior of our algorithm [E1]. This design choice allows the experiments to clearly demonstrate the effectiveness of learning constraints and the advantages of the bi-level formulation. We reran the experiment with 20% incorrect feedback, and the results are shown in the global response.
>
> [E1] Lee, Kimin, et al. "Pebble: Feedback-efficient interactive reinforcement learning via relabeling experience and unsupervised pre-training." arXiv preprint arXiv:2106.05091 (2021).
>
> >Missing Key Baseline: The paper fails to compare against the most obvious baseline: a standard, iterative RLHF approach where the update cycle is simply run much more frequently. It is unclear if the proposed complex optimization is better than just iterating faster.
>
> **Answer:** We thank the reviewer for identifying this baseline. We conducted an additional experiment by running PEBBLE with a more frequent update cycle on the Walker2D and HalfCheetah environments. In our original PEBBLE baseline, the reward function is updated every 2000 environment steps. In the more frequent version, we update the reward function every 1000 environment steps. The cumulative reward results of the two PEBBLE baselines and the PARL baseline, evaluated over 1 million environmental steps with the maximum trajectory length set to 200, are shown in the following table.
>
> | Environment   | Original One        | More Frequent One     | PARL               |
> |---------------|----------------------|------------------------|--------------------|
> | Walker2d      | 510.86 ± 16.29       | 521.20 ± 15.67         | 530.18 ± 12.13     |
> | HalfCheetah   | 1065.49 ± 36.78      | 1136.52 ± 30.83        | 1546.75 ± 50.56    |
>
>
> The results show that using a more frequent update cycle yields a modest improvement in cumulative reward for both environments. However, the gains remain substantially smaller than those achieved by PARL, which benefits from the bi-level framework. Notably, in the HalfCheetah environment, the performance gap between PARL and both PEBBLE variants is particularly large. In conclusion, while increasing the update frequency slightly improves PEBBLE’s performance, the improvement is limited, and the bi-level formulation used in PARL provides significantly greater benefits.
>
> >Lack of Ablation Study: The method introduces two key components (a cost function and a bi-level framework) but provides no ablation study to disentangle their effects. We cannot know what is truly responsible for the performance changes.
>
> **Answer:** There are comparisons for each individual component of our approach (the cost function and the bi-level framework). We include three baselines: PEBBLE, PARL, and Safe RLHF. PEBBLE serves as the naïve RLHF baseline without a cost function and without a bi-level framework. PARL includes a bi-level framework but does not use a cost function. Safe RLHF incorporates a cost function but does not use a bi-level framework. The performance difference between PEBBLE and PARL highlights the effect of introducing the bi-level framework, while the performance difference between PEBBLE and Safe RLHF reveals the effect of adding a cost function.
>
> >Lack of Analysis: The paper presents results but offers almost no analysis. It claims success even when the method underperforms baselines on some metrics (e.g., Hopper return), with no discussion as to why.
>
> **Answer:** Please refer to the response in the global response.

---

> ### Author Response · Authors · 2025-11-28
>
> >Q1:Could the authors clarify how strong duality (Lemma 1) is guaranteed to hold for the problems?
>
> **Answer:** We thank the reviewer for this question. Lemma 1 relies on including the entropy term $H(\pi)$ in the lower-level objective function. The entropy term establishes a relationship between $\pi$ and $\lambda$, ensuring that $\nabla_\lambda \pi$ exists.  The Lagrangian of this primal problem is $L(\pi;\phi,\psi,\lambda) \triangleq J_{r_{\phi}}(\pi)+H(\pi) - \lambda J_{c_{\psi}}(\pi) $. By taking the derivative of $L(\pi;\phi,\psi,\lambda)$ with respect to $\pi$, we obtain $\frac{\partial L(\pi;\phi,\psi,\lambda)}{\partial \pi(a|s)}=P(s_h=s)\gamma^h(1-H(\pi,s,a)+J_r(\pi,s,a)-\lambda
>         J_c(\pi,s,a))$ where $P(s_h=s)$ denotes the probability of visiting state $s$ at time step $h$. At the maximizer of the Lagrangian, the first-order optimality condition implies $\frac{\partial L(\pi;\phi,\psi,\lambda)}{\partial \pi(a|s)} =0$. Then, we can get $1-H(\pi_{\phi, \psi,\lambda(\phi,\psi)},s,a)+J_r(\pi_{\phi, \psi,\lambda(\phi,\psi)},s,a)-\lambda
>         J_c(\pi_{\phi, \psi,\lambda(\phi,\psi)},s,a) = 0$.
>
> The dual function is $\min_{\lambda\geq 0}G(\lambda; \phi,\psi )$, where $G(\lambda; \phi,\psi ) \triangleq \max_{\pi} J_{r_{\phi}}(\pi)+H(\pi) - \lambda J_{c_{\psi}}(\pi) = J_{r_{\phi}}(\pi_{\phi, \psi,\lambda(\phi,\psi)})+H(\pi_{\phi, \psi,\lambda(\phi,\psi)}) - \lambda J_{c_{\psi}}(\pi_{\phi, \psi,\lambda(\phi,\psi)})$. By calculating the gradient of $G(\lambda; \phi,\psi )$ with respect to $\lambda$, we can get $\nabla_{\lambda}G(\lambda;\phi,\psi) = \int_{s\in S} f(\pi_{\phi, \psi,\lambda(\phi,\psi)},s)\int_{a\in A} \pi_{\phi, \psi,\lambda(\phi,\psi)}(a|s)\nabla_{\lambda}\ln(\pi_{\phi, \psi,\lambda(\phi,\psi)}(a|s)) (1-H(\pi_{\phi, \psi,\lambda(\phi,\psi)},s,a)+ J_{r_{\phi}}(\pi_{\phi, \psi,\lambda(\phi,\psi)},s,a)
>         - \lambda J_{c_{\psi}}(\pi_{\phi, \psi,\lambda(\phi,\psi)},s,a))dads - J_{c_{\psi}}(\pi_{\phi, \psi,\lambda(\phi,\psi)})$, where $f(\pi_{\phi,\psi,\lambda},s)$ is the state visitation frequency. Since we have $1-H(\pi_{\phi, \psi,\lambda(\phi,\psi)},s,a)+J_r(\pi_{\phi, \psi,\lambda(\phi,\psi)},s,a)-\lambda
>         J_c(\pi_{\phi, \psi,\lambda(\phi,\psi)},s,a) = 0$, then we can get $\nabla_{\lambda}G(\lambda;\phi,\psi) = - J_{c_{\psi}}(\pi_{\phi, \psi,\lambda(\phi,\psi)})$. When $G(\lambda;\phi,\psi)$ reach the minimum, the gradient $\nabla_{\lambda}G(\lambda;\phi,\psi)$ should be $0$ which means $J_{c_{\psi}}(\pi_{\phi,\psi,\lambda^{\star}(\phi,\psi)}) = 0$. We can see that $\pi_{\phi, \psi,\lambda^{\star}(\phi,\psi)}$ is a feasible solution for the primal constrained RL problem.
>
> Consider $d^{\star}$ as the optimal value of the dual problem $min_{\lambda\geq 0}G(\lambda; \phi,\psi )$ and $p^{\star}$ as the optimal value of the primal problem. Then, we can get $H(\pi_{\phi, \psi,\lambda^{\star}(\phi,\psi)}) + J_{r_{\phi}}(\pi_{\phi, \psi,\lambda^{\star}(\phi,\psi)})\leq p^{\star} \leq d^{\star} = G(\lambda^{\star}(\phi,\psi);\phi,\psi) = H(\pi_{\phi, \psi,\lambda^{\star}(\phi,\psi)}) + J_{r_{\phi}}(\pi_{\phi, \psi,\lambda^{\star}(\phi,\psi)})$, shows that the strong duality holds.

---

### Official Review · Reviewer_b4zk · 2025-10-29

**Soundness:** 2
**Presentation:** 2
**Contribution:** 3
**Rating:** 4
**Confidence:** 3

**Summary:**

This paper proposes a bi-level optimization framework for RLHF. The upper-level optimization learns both a reward and a cost function from preference feedback, while the lower-level optimization solves the dual of a constrained RL problem. To handle the non-smoothness of the objective, the authors employ the Clarke subdifferential framework to approximate the hypergradient and provide convergence and sample complexity analyses. Empirical results demonstrate that the proposed algorithm outperforms three baselines across four environments.

**Strengths:**

- The integration of a bi-level optimization framework into the constrained RLHF setting is well-motivated and appears novel.
- Although I did not check the proofs in full detail, the proposed method for addressing the non-convex lower-level optimization problem seems mathematically sound and appropriate.

**Weaknesses:**

- Some notations, particularly those related to the cost function, are unclear (see questions below). I also recommend the authors carefully proofread the Appendix. Several proofs are difficult to follow and contain typos, which hinder readability.
- The experimental results are not sufficiently comprehensive to support the main claims. Including more environments or additional ablation studies would strengthen the empirical evidence.

**Questions:**

- My understanding is that a high value of $J_c (\tau)$ indicates a trajectory with a high cumulative cost (i.e., a worse trajectory). Then, In line 197, why does the BT model seem to prefer trajectories with higher costs? Furthermore, in Theorem 2, why is the negative cost of the learned policy upper-bounded?
- In lines 441–444, when generating the synthetic feedback data, is a greedy model used instead of the BT model? If so, could the authors clarify the rationale?
- Minor typos: (i) In line 401, $\epsilon$ should be $\epsilon_k$. (ii) In line 416, $(\pi_)$ should be $(\pi_h)$.

---

> ### Author Response · Authors · 2025-11-28
>
> >W1: Some notations, particularly those related to the cost function, are unclear (see questions below). I also recommend the authors carefully proofread the Appendix. Several proofs are difficult to follow and contain typos, which hinder readability.
>
> **Answer:** We thank the reviewer for this valuable feedback. We will clarify all cost-related notations, thoroughly proofread the appendix to remove remaining typos, and revise the proofs to improve readability.
>
> >W2: The experimental results are not sufficiently comprehensive to support the main claims. Including more environments or additional ablation studies would strengthen the empirical evidence.
>
> **Answer:** We thank the reviewer for this constructive suggestion. We will adapt our algorithm for large-scale RLHF and LLM experiments. Specifically, we will modify the method to a single-loop formulation in which the inner-level variable is updated once per outer iteration, thereby avoiding repeated inner-loop optimization. In addition, we will use first-order gradient approximations in place of second-order derivatives to reduce the computational overhead associated with Hessian calculations. These improvements will be incorporated and clearly explained in the revised paper.
>
> >Q1: My understanding is that a high value of $J_c(\tau)$ indicates a trajectory with a high cumulative cost (i.e., a worse trajectory). Then, in line 197, why does the BT model seem to prefer trajectories with higher costs? Furthermore, in Theorem 2, why is the negative cost of the learned policy upper-bounded?
>
> **Answer:** Thank you for the question. The BT model in line 197 is used to illustrate how human choices are predicted from the current reward and cost functions. As the reviewer correctly noted, if viewed in isolation, the BT model assigns a higher selection probability to the trajectory with a higher cumulative cost, since BT prefers trajectories with a higher input score. However, in our formulation, this behavior is intentionally counteracted in the loss: the preference label $y_c$ ensures that trajectories judged as “worse” by the human are assigned higher cumulative cost values during learning. Thus, the BT model serves only as a likelihood model for preferences, and the learning dynamics push the cost function in the correct direction.
>
> Regarding Theorem 2, the sub-optimality gap for the cost should be expressed in absolute value since $SubOptC(\pi_{\phi^{\star},\psi^{\star},\lambda^{\star}(\phi^{\star},\psi^{\star})})\triangleq J_{c_{h}}(\pi_h)-J_{c_{h}}(\pi_{\phi^{\star},\psi^{\star},\lambda^{\star}(\phi^{\star},\psi^{\star})})\leq 0$. We change it to $|SubOptC(\pi_{\phi^{\star},\psi^{\star},\lambda^{\star}(\phi^{\star},\psi^{\star})})| \leq \mathcal{O}(\sqrt{\frac{1}{Kn}\ln(\frac{ N(\mathcal{F}, \|\cdot\|_{\infty},\frac{1}{Kn})}{\delta})})$
>
> >Q2: In lines 441–444, when generating the synthetic feedback data, is a greedy model used instead of the BT model? If so, could the authors clarify the rationale?
>
> **Answer:** We appreciate the reviewer’s question. Yes, we use a greedy model to generate the synthetic preference data. As mention in [D1], noise-free labels allow us to isolate and directly evaluate the optimization behavior of the algorithm. In our experiments, we aim to clearly demonstrate the effectiveness of learning constraints and the advantages of the bi-level formulation.
> If we were to use the BT model to generate feedback, the injected stochasticity would obscure these algorithmic effects and make it more difficult to attribute performance changes to the method itself. Therefore, we use a greedy model as a deterministic oracle, which provides clean and unambiguous preference labels, enabling a clearer evaluation of the optimization dynamics. We reran the experiment with 20% incorrect feedback, and the results are shown in the global response.
>
> [D1] Lee, Kimin, et al. "Pebble: Feedback-efficient interactive reinforcement learning via relabeling experience and unsupervised pre-training." arXiv preprint arXiv:2106.05091 (2021).

---

### Official Review · Reviewer_Ccqt · 2025-10-30

**Soundness:** 2
**Presentation:** 2
**Contribution:** 2
**Rating:** 4
**Confidence:** 3

**Summary:**

This paper studies the problem of jointly learning a reward function, a cost function, and a policy from human feedback. The authors formulate the problem as a constrained bi-level optimization problem, where the upper level infers the reward and cost functions from feedback, while the lower level optimizes a policy to best align with that feedback. To solve this problem, the authors propose a double-loop algorithm, Constrained Bi-level Optimization for Reinforcement Learning from Human Feedback (CB-RLHF), which solves the lower-level optimization problem in the inner loop and the upper-level optimization problem in the outer loop. The authors establish a theoretical guarantee that CB-RLHF converges at a rate of $O(1/\sqrt{K})$, and demonstrate its effectiveness across multiple simulation environments.

**Strengths:**

1. The studied problem, constrained reinforcement learning from human feedback (RLHF), is well-motivated and finds important applications in large language model alignment.
2. The idea of formulating the constrained RLHF problem as a constrained bi-level optimization problem is interesting.
3. The authors propose a double-loop algorithm and provide a theoretical guarantee on convergence.

**Weaknesses:**

1. The writing and readability of this paper needs to be improved.
2. In Eq. (1), what is the motivation of using $H(\pi)$ as a regularization term, instead of the KL divergence with the reference policy?
3. It is hard to understand the theoretical results provided in this paper. (i) The abstract of this paper mentioned that they prove that algorithm CB-RLHF converges at a rate of $O(1/\sqrt{K})$. However, Theorem 1 only states that the gradient is bounded by $O(1/\sqrt{K})$. How does this result imply the convergence rate to the globally optimal policy?
(ii) In Theorem 2, what are the definitions of SubOptR and SubOptC, in particular, what is the definition of human policy $\pi_h$? Does Theorem 2 provide the performance gap between the optimal policy and the output policy of algorithm CB-RLHF? If it is, why is it not dependent on $K$?
4. The constrained LLM alignment problem has been studied in several prior works. The authors should discuss more on the advantages of the proposed constrained bi-level optimization approach compared to the existing primal-dual approaches, e.g., Safe RLHF [Dai et al., 2023].
5. This paper only provides experiments on MuJoCo. It would enhance this paper if the authors can provide experiments on LLMs.

**Questions:**

Please see the weaknesses above.

---

> ### Author Response · Authors · 2025-11-28
>
> >W1: The writing and readability of this paper needs to be improved.
>
> **Answer:** We appreciate the reviewer’s suggestion. We will revise the paper to significantly improve clarity and readability, including refining notation, reorganizing sections for better flow, and improving the explanations in both the main text and the appendix.
>
> >W2: In Eq. (1), what is the motivation of using $H(\pi)$ as a regularization term, instead of the KL divergence with the reference policy?
>
> **Answer:** We appreciate the reviewer’s question. With the entropy regularize $H(\pi)$, the optimal policy takes the soft Bellman form   $\pi_{\phi, \psi, \lambda}(a|s) = \frac{\exp (Q^{soft}_{\phi, \psi,\lambda}(s,a))}{\exp (V^{soft}\_{\phi, \psi, \lambda}(s))}$, where $Q^{soft}\_{\phi, \psi,\lambda} (s,a) =  r\_{\phi}(s,a) - \lambda c\_{\psi}(s,a) + \gamma \int\_{s'\in S} P(s'| s,a) V^{soft}\_{\phi, \psi,\lambda}(s') ds'$, and $V^{soft}\_{\phi, \psi,\lambda}(s) = \ln \int\_{a \in A} \exp (Q^{soft}\_{\phi, \psi,\lambda} (s,a)) da.$ This property allows us to obtain a closed-form relationship between the reward function, the cost function, and the resulting optimal policy, which is $\pi\_{\phi, \psi, \lambda}(a|s) = \frac{\exp (Q^{soft}\_{\phi, \psi,\lambda}(s,a))}{\exp (V^{soft}\_{\phi, \psi, \lambda}(s))}$.  This relationship enables efficient and tractable computation of gradients with respect to the reward and cost parameters ($\psi,\phi$). Since our problem setting does not assume the availability of a reference policy, we use the entropy regularizer $H(\pi)$ rather than a KL-divergence term.
>
> >W3:It is hard to understand the theoretical results provided in this paper. (i) The abstract of this paper mentioned that they prove that the algorithm CB-RLHF converges at a rate of $\mathcal{O}(\frac{1}{\sqrt{K}})$. However, Theorem 1 only states that the gradient is bounded by $\mathcal{O}(\frac{1}{\sqrt{K}})$. How does this result imply the convergence rate to the globally optimal policy? (ii) In Theorem 2, what are the definitions of SubOptR and SubOptC, in particular, what is the definition of human policy $\pi_h$ ? Does Theorem 2 provide the performance gap between the optimal policy and the output policy of algorithm CB-RLHF? If it is, why is it not dependent on $K$?
>
> **Answer:** Thank you for the question. (i) Because the constrained bi-level optimization problem (6) is non-convex, we can only guarantee convergence to a stationary point, rather than a global optimum. Establishing convergence through the upper-level hypergradient is a standard approach in bi-level optimization (e.g., [C1, C2, C3]). When the hypergradient converges, the learned reward and cost parameters ($\psi,\phi$) converge to stationary points ($\psi^{\star},\phi^{\star}$), and the policy converges to the corresponding solution $\pi_{\phi^{\star}, \psi^{\star},\lambda^{\star}(\phi^{\star},\psi^{\star})}$.
>
> (ii) We assume the human labels are generated according to the ground-truth reward function $r_h$ and the ground-truth cost function $c_h$. The human policy $\pi_h$ is the optimal solution of the constrained RL problem where $r_h$ is the reward function and $c_h$ is the cost function. We define the cumulative reward difference between $\pi_{\phi^{\star},\psi^{\star},\lambda^{\star}(\phi^{\star},\psi^{\star})}$ and human policy $\pi_h$ according to the true reward function $r_h$ as $SubOptR(\pi_{\phi^{\star},\psi^{\star},\lambda^{\star}(\phi^{\star},\psi^{\star})}) \triangleq J_{r_{h}}(\pi_h)-J_{r_{h}}(\pi_{\phi^{\star},\psi^{\star},\lambda^{\star}(\phi^{\star},\psi^{\star})})$. Analogously, the cumulative cost difference between $\pi_{\phi^{\star},\psi^{\star},\lambda^{\star}(\phi^{\star},\psi^{\star})}$ and $\pi_h$ according to the true cost function $c_h$ is defined as $SubOptC(\pi_{\phi^{\star},\psi^{\star},\lambda^{\star}(\phi^{\star},\psi^{\star})})\triangleq |J_{c_{h}}(\pi_h)-J_{c_{h}}(\pi_{\phi^{\star},\psi^{\star},\lambda^{\star}(\phi^{\star},\psi^{\star})})|$. Yes, Theorem 2 provides the performance gap between the optimal policy and the policy returned by CB-RLHF. We define $n$ as the number of preference data collected per outer iteration. Since the total number of outer-loop iterations is $K$, we consider whole data used and the result becomes $\mathcal{O}(\sqrt{\frac{1}{Kn}\ln(\frac{ N(\mathcal{F}, \|\cdot\|_{\infty},\frac{1}{Kn})}{\delta})})$. We thank the reviewer for pointing this out.
>
>
> [C1]Ghadimi, Saeed, et al. "Approximation methods for bilevel programming." arXiv preprint arXiv:1802.02246 (2018).
>
> [C2]Zeng, Siliang, et al. "Maximum-likelihood inverse reinforcement learning with finite-time guarantees." Advances in Neural Information Processing Systems 35 (2022): 10122-10135.
>
> [C3]Liu, Shicheng, et al. "Distributed inverse constrained reinforcement learning for multi-agent systems." Advances in Neural Information Processing Systems 35 (2022): 33444-33456.

---

> > ### Author Response · Authors · 2025-11-28
> >
> > > W4: The constrained LLM alignment problem has been studied in several prior works. The authors should discuss more on the advantages of the proposed constrained bi-level optimization approach compared to the existing primal-dual approaches, e.g., Safe RLHF [Dai et al., 2023].
> >
> > **Answer:** We thank the reviewer for this suggestion, and we highlight the key differences below. As described in lines 57–65, Safe RLHF performs feedback collection, reward and cost learning, and policy optimization in a sequential and decoupled manner. As a result, the reward function is learned from feedback generated by an earlier policy, while the policy continues to evolve. This decoupling overlooks the dependence of the learned reward on the changing data distribution induced by the policy. Consequently, Safe RLHF may suffer from misalignment, where the learned reward no longer reflects the human’s true intent, potentially leading to suboptimal policies.
> >
> > In contrast, our method explicitly addresses this limitation through a constrained bi-level formulation that tightly couples the upper-level (reward and cost learning) and lower-level (policy optimization) problems. The human feedback used to update the reward and cost functions is always generated from trajectories induced by the current optimal policy associated with the current parameters. This alignment ensures that the learned reward and cost functions remain consistent with human preference throughout training, mitigating the distribution shift issues inherent in Safe RLHF. As shown in the experimental results presented in the global response, methods incorporating a bi-level framework (PARL and ours) attain higher cumulative rewards compared to Safe RLHF.
> >
> > > W5: This paper only provides experiments on MuJoCo. It would enhance this paper if the authors can provide experiments on LLMs.
> >
> > **Answer:** We thank the reviewer for this suggestion. We fully agree that evaluating CB-RLHF on LLMs would further strengthen the paper. To improve the efficiency of our method in large-scale LLM experiments, we will incorporate the following ideas. First, we will update the inner-level variable only once per outer iteration, thereby avoiding repeated inner-loop optimization and reducing the algorithm to a single-loop format. Second, we will employ first-order gradient estimators in place of second-order derivatives, substantially lowering the computational cost. These modifications can significantly improve scalability and make CB-RLHF more suitable for LLM settings. We will add these discussions to the revised paper.

---

### Official Review · Reviewer_bG2X · 2025-11-01

**Soundness:** 2
**Presentation:** 3
**Contribution:** 2
**Rating:** 2
**Confidence:** 3

**Summary:**

This paper proposes CB-RLHF, a bi-level optimization framework for solving constrained RLHF problems.
1. The upper level learns reward and cost functions from human feedback, while the lower level solves a dual-convex reformulation of the constrained RL problem using the Lagrangian dual.
2. The method employs Clarke subdifferential and gradient approximation to handle non-differentiable hypergradients and proves a convergence rate of O(1/√K) and an approximation error of O(1/√N).
3. The algorithm is compared against PEBBLE, Safe-RLHF, and PARL on four MuJoCo tasks, showing that CB-RLHF improves constraint satisfaction and return performance.

**Strengths:**

1. The theoretical formulation appears rigorous and well-grounded, even though I didn't fully verify all the proof details.
2. The paper provides a complete methodological pipeline—from problem formulation to convergence theory and experiments—with a coherent narrative.
3. The combination of theory and empirical validation makes the proposed framework conceptually convincing.

**Weaknesses:**

1. The method critically relies on strong duality to convert a non-convex constrained problem into a convex one, yet the paper does not discuss or verify the duality gap.
   In practical RLHF settings involving approximation and sampling, zero duality gap is unlikely to hold. The authors should analyze or empirically estimate the duality gap’s impact on convergence and correctness, or at least clarify conditions under which strong duality is justified.

2. The bi-level structure is theoretically elegant but computationally heavy. Frequent estimation of λ*(ϕ, ψ) and inner-loop optimization make the algorithm expensive.
   Its feasibility for large-scale RLHF—especially in LLM fine-tuning scenarios—remains untested. The paper would benefit from demonstrating CB-RLHF on a small-scale language model (e.g., 1.5B parameters) to establish practical viability.

3. The experimental design mixes reward and cost by defining return = reward − cost, which is inconsistent with the problem formulation that treats reward and cost separately.
   Because reward and cost are in different units and the dual algorithm is scale-insensitive (replacing λ, c with αλ, c/α does not dramatically change results), this modification obscures interpretability. It would be more principled to report cumulative reward and constraint violation separately.

4. Experimental fairness is questionable. The total training steps vary across the four environments, and results seem sensitive to stopping criteria.
   For instance, in Walker2D, using 1e5 steps (like HalfCheetah and Swimmer) may allow PARL to outperform CB-RLHF; conversely, reducing HalfCheetah to 0.6e5 steps (like Hopper) makes CB-RLHF weaker in both return and constraint violation. This suggests possible cherry-picking of stopping points.

5. The proposed framework effectively assumes a Bradley–Terry preference model. As such, its applicability is limited to preference-based feedback rather than general RLHF (which can involve scalar rewards, rankings, or textual feedback).
   The title should more accurately read “A Constrained Bi-level Framework for Preference-based Human Feedback” rather than “RLHF,” which implies broader generality.

6. Minor issue: in line 212, z₀ and z₁ should be s₀ and s₁.

**Questions:**

1. How is the number of inner-loop dual updates tₖ determined? Is it adaptive, fixed, or tuned per environment?
2. Can the authors provide theoretical guarantees or empirical analysis under small (non-zero) duality gaps? What happens if strong duality fails slightly—does convergence degrade gracefully?

---

> ### Author Response · Authors · 2025-11-28
>
> > W1: The method critically relies on strong duality to convert a non-convex constrained problem into a convex one, yet the paper does not discuss or verify the duality gap. In practical RLHF settings involving approximation and sampling, zero duality gap is unlikely to hold. The authors should analyze or empirically estimate the duality gap’s impact on convergence and correctness, or at least clarify conditions under which strong duality is justified.
>
> **Answer:**  We thank the reviewer for raising this important point. Lemma 1 establishes strong duality under the assumption that the expectations are computed exactly. We agree that, in practice, these expectations are estimated from samples, which can introduce a duality gap. When such a gap exists, the primal and dual solutions no longer coincide, resulting in a discrepancy between the learned $\hat{\lambda}$ and the optimal solution $\lambda^{\star}$. Next, we will theoretically analyze how this discrepancy affects the convergence guarantee. Suppose this discrepancy is bounded by a constant $\epsilon_d>0$,i.e., $\|\hat{\lambda}-\lambda^{\star}\| \leq \epsilon_d$. In Appendix A.7, we have proved that $\|H(\phi_1,\psi,\hat{\lambda})-H(\phi_2,\psi,\lambda^{\star})\|\leq \| \hat{\lambda}-\lambda^{\star}\|$. Therefore, the discrepancy leads to $\|H(\phi_1,\psi,\hat{\lambda})-H(\phi_2,\psi,\lambda^{\star})\|\leq \epsilon_d$. Thus, an additional hypergradient error $\epsilon_d$ needs to be considered in the convergence analysis. Following the same proof steps in Appendix A.8, the additional hypergradient error $\epsilon_d$ leads to an additional term $\mathcal{O}(\frac{\epsilon_d^2}{\sqrt{K}})$ in the final convergence rate result. However, this additional term does not change the dominant term in the convergence bound. The overall convergence rate of the bi-level algorithm therefore remains $\mathcal{O}(\frac{1}{\sqrt{K}})$, and the factor $\epsilon_d^2$ arising from the approximation error is absorbed into the multiplicative constant of the $\frac{1}{\sqrt{K}}$ rate.
>
> >W2: The bi-level structure is theoretically elegant but computationally heavy. Frequent estimation of $\lambda^*(\phi,\psi)$ and inner-loop optimization make the algorithm expensive.
> Its feasibility for large-scale RLHF—especially in LLM fine-tuning scenarios—remains untested. The paper would benefit from demonstrating CB-RLHF on a small-scale language model (e.g., 1.5B parameters) to establish practical viability.
>
> **Answer:**  We appreciate the reviewer for raising this important practical concern. Recent work has demonstrated that bi-level optimization can indeed scale to large language models. For example, [A1] reports experiments with a 1.3B-parameter model, while [A2] includes results on a 32B-parameter model. In large-scale settings (including LLM fine-tuning), the bi-level algorithm can be adapted to a single-loop formulation and equipped with a first-order hypergradient estimator [A2, A3]. In the single-loop formulation, the inner-level variable $\lambda^{\star}(\phi,\psi)$ is updated once per outer iteration, avoiding repeated inner-loop optimization. The first-order estimator further reduces computational cost by bypassing second-order derivative calculations. Together, these modifications substantially improve scalability and make bi-level methods feasible for large-scale RLHF and LLM applications.  The discussion above on efficiency techniques for making the bi-level formulation practical in large-scale experiments will be added to the Appendix for completeness and clarity.
>
> [A1] Yu, Yang, et al. "LLM Data Selection and Utilization via Dynamic Bi-level Optimization." arXiv preprint arXiv:2507.16178 (2025).
>
> [A2]Pan, Rui, et al. "Scalebio: Scalable bilevel optimization for llm data reweighting." Proceedings of the 63rd Annual Meeting of the Association for Computational Linguistics (Volume 1: Long Papers). 2025.
>
> [A3]Kwon, Jeongyeol, et al. "A fully first-order method for stochastic bilevel optimization." International Conference on Machine Learning. PMLR, 2023.
>
> > W3: The experimental design mixes reward and cost by defining return = reward $-$ cost, which is inconsistent with the problem formulation that treats reward and cost separately. Because reward and cost are in different units and the dual algorithm is scale-insensitive, this modification obscures interpretability. It would be more principled to report cumulative reward and constraint violation separately.
>
> **Answer:** Please refer to the response in the global response.

---

> > ### Author Response · Authors · 2025-11-28
> >
> > >W4: Experimental fairness is questionable. The total training steps vary across the four environments, and results seem sensitive to stopping criteria. For instance, in Walker2D, using 1e5 steps (like HalfCheetah and Swimmer) may allow PARL to outperform CB-RLHF; conversely, reducing HalfCheetah to 0.6e5 steps (like Hopper) makes CB-RLHF weaker in both return and constraint violation. This suggests possible cherry-picking of stopping points.
> >
> > **Answer:** We reran the experiment with 1 million environment steps. Please refer to the global response for details.
> >
> > >W5: The proposed framework effectively assumes a Bradley–Terry preference model. As such, its applicability is limited to preference-based feedback rather than general RLHF (which can involve scalar rewards, rankings, or textual feedback). The title should more accurately read “A Constrained Bi-level Framework for Preference-based Human Feedback” rather than “RLHF,” which implies broader generality.
> >
> > **Answer:**  Thank you for the comment. We will revise the title to ''A Constrained Bi-level Framework for Preference-based Human Feedback''.
> >
> > >W6:Minor issue: in line 212, $z_0$ and $z_1$ should be $s_0$ and $s_1$.
> >
> > **Answer:** Thank you for pointing this out. We will address this by replacing  $s_0$ and $s_1$ with $z_0$ and $z_1$, consistent with our notation for classification labels.
> >
> > > Q1:How is the number of inner-loop dual updates $t_k$ determined? Is it adaptive, fixed, or tuned per environment?
> >
> > **Answer:** Thank you for the question. In the convergence analysis, we require an upper bound on the sum $\sum_{k=0}^{K-1} \eta^{2t_k}, $ where $\eta\in(0,1)$. Following the guidance of [B1], we choose an adaptive number of inner-loop iterations $t_k = \lceil \frac{\sqrt[4]{k+1}}{2}\rceil$. With this choice, it can be shown that $\sum_{k=0}^{K-1} \eta^{2t_k} \leq \frac{15\eta}{(1-\eta)^4}$. This value $\lceil \frac{\sqrt[4]{k+1}}{2}\rceil$ balances the trade-off between computational cost (minimizing inner loop iteration number) and approximation accuracy (reducing the error sufficiently fast), enabling efficient and stable convergence of the bi-level algorithm.
> >
> > [B1] Ghadimi, Saeed, et al. "Approximation methods for bilevel programming." arXiv preprint arXiv:1802.02246 (2018).
> >
> > >Q2: Can the authors provide theoretical guarantees or empirical analysis under small (non-zero) duality gaps? What happens if strong duality fails slightly—does convergence degrade gracefully?
> >
> > **Answer:** Please refer to the answer for Weakness 1.

---

### Author Response · Authors · 2025-11-28

We sincerely thank all reviewers for their thoughtful comments and constructive feedback. We reran the experiments on the Walker, HalfCheetah, and Ant environments using $1$ million environment steps and including
20% incorrect feedback. The resulting cumulative rewards, cumulative costs, and constraint-violation rates are presented in the tables below.

The cumulative reward:

| Environment   | PEBBLE             | PARL               | Safe RLHF          | Ours               |
|---------------|--------------------|--------------------|--------------------|--------------------|
| Walker        | 510.86 ± 16.29     | 530.18 ± 12.13     | 517.24 ± 23.90     | 586.69 ± 24.56     |
| HalfCheetah   | 1065.49 ± 36.78    | 1546.75 ± 50.46    | 1034.50 ± 10.74    | 1753.57 ± 25.97    |
| Ant           | 561.82 ± 28.22     | 697.43 ± 34.11     | 583.13 ± 42.83     | 761.23 ± 51.64     |


The cumulative cost:

| Environment   | PEBBLE            | PARL              | Safe RLHF         | Ours              |
|---------------|-------------------|-------------------|-------------------|-------------------|
| Walker        | 18.05 ± 2.92      | 25.87 ± 8.83      | 5.99 ± 2.18       | 6.64 ± 5.98       |
| HalfCheetah   | 91.82 ± 7.55      | 153.58 ± 4.39     | 23.95 ± 3.80      | 28.88 ± 1.98      |
| Ant           | 107.21 ± 16.61    | 90.61 ± 4.49      | 9.89 ± 1.05       | 9.44 ± 0.70       |


The constraint violation rate (reported as a percentage):

| Environment   | PEBBLE            | PARL              | Safe RLHF         | Ours              |
|---------------|-------------------|-------------------|-------------------|-------------------|
| Walker        | 83.02 ± 1.45     | 82.98 ± 2.97     | 23.30 ± 1.94     | 25.35 ± 1.24    |
| HalfCheetah   | 98.42 ± 1.26     | 98.79 ± 0.99     | 19.87 ± 1.32     | 19.86 ± 1.57     |
| Ant           | 96.29 ± 2.19     | 94.10 ± 2.19     | 32.55 ± 2.59     | 35.37 ± 5.22     |

First, from the cumulative reward results, we observe that algorithms equipped with a bi-level framework (PARL and ours) achieve higher cumulative rewards than those without one (PEBBLE and Safe RLHF). This indicates that the bi-level structure leads to a policy that is more aligned with the ground-truth optimal policy in terms of reward maximization.

Second, the cumulative cost results show that algorithms that learn a cost function (Safe RLHF and ours) attain lower cumulative costs and lower constraint-violation rates than those without learning a cost function (PEBBLE and PARL). This demonstrates that learning a cost function contributes to a safer policy with fewer constraint violations.

In conclusion, our algorithm effectively solves the constrained RLHF problem, achieving strong performance while ensuring constraint satisfaction and simultaneously benefiting from the bi-level framework and cost-function learning.

---

### Meta-Review · Area_Chair_QDyf · 2025-12-26

**Summary:**

Reviewers generally agreed that the paper addresses an important and timely problem in constrained reinforcement learning from human feedback, and that framing it as a constrained bi-level optimization problem is conceptually appealing. Some key concerns include strong duality, complexity and scalability, experimental design and fairness, and scope and positioning of the paper. The method assumes a Bradley–Terry preference model, which however is insensitive to mean shift of the cost function, and therefore is not suitable for constrianed RL problem which compares the cost value function with a given threshold.

**Reviewer Concerns:**

There are some concerns about the experimental results, which were adequately addressed during the rebuttal. The scalability issue was not fully addressed in the rebuttal.

**Reviewer Scores:**

Reviewer bG2X: 2->2
Reviewer Ccqt: 4->4
Reviewer b4zk: 4->4
Reviewer mhU2: 4->4

---

### Decision · Program_Chairs · 2026-01-26

Reject